# Investigating the role of group-based morality in extreme behavioral expressions of prejudice

Joe Hoover [1,2], Mohammad Atari [1,2], Aida Mostafazadeh Davani[2,3,4], Brendan Kennedy[2,3,4], Gwenyth Portillo-Wightman[1], Leigh Yeh [3] & Morteza Dehghani [1,2,3✉]

Understanding motivations underlying acts of hatred are essential for developing strategies to prevent such extreme behavioral expressions of prejudice (EBEPs) against marginalized groups. In this work, we investigate the motivations underlying EBEPs as a function of moral values. Specifically, we propose EBEPs may often be best understood as morally motivated behaviors grounded in people's moral values and perceptions of moral violations. As evidence, we report five studies that integrate spatial modeling and experimental methods to investigate the relationship between moral values and EBEPs. Our results, from these U.S. based studies, suggest that moral values oriented around group preservation are predictive of the county-level prevalence of hate groups and associated with the belief that extreme behavioral expressions of prejudice against marginalized groups are justified. Additional analyses suggest that the association between group-based moral values and EBEPs against outgroups can be partly explained by the belief that these groups have done something morally wrong.

[1] Department of Psychology, University of Southern California, Los Angeles, CA, USA. [2] Brain and Creativity Institute, University of Southern California, Los Angeles, CA, USA. [3] Department of Computer Science, University of Southern California, Los Angeles, CA, USA. [4]These authors contributed equally: Aida Mostafazadeh Davani, Brendan Kennedy. ✉email: mdehghan@usc.edu

Throughout history, humans have discriminated against and persecuted other humans because of their identities[1–3]. Such acts of hatred have detrimental effects on survivors and survivors' communities. Survivors of hate crime, for example, experience higher levels of depression and anxiety compared to survivors of comparable crimes not motivated by bias[4] and they may ultimately reject or despise the part of their identity that was targeted[5]. Even for people who are not directly targeted by abusers, sharing a trait targeted by a hate crime can cause clinical levels of posttraumatic stress[6] and elevated levels of depression and anxiety in survivors[7].

The human tendency toward identity-based hatred and violence remains a major contributor to human suffering. While some have argued that there have been major long-term decreases in global levels of violence and prejudice[8], the number of reported hate crimes in the United States increased by 12.5% in 2017, the fourth consecutive year with a positive trend, while general crime rates experienced a decline during the same period[9,10]. In 2020 and 2021, in the wake of the COVID-19 pandemic, hate crimes targeting Asian Americans spiked substantially in the US[11]. In addition, many European countries have seen increased animosity toward marginalized groups—from citizens as well as political parties—amidst the influx of refugees from countries affected by the armed conflict during 2015–2017[12] and during the COVID-19 pandemic[13]. This trend is particularly concerning since some crimes motivated by hate are unreported or underreported by local authorities[14]. The number of hate groups operating in the U.S. has also recently reached a record high according to the Southern Poverty Law Center (SPLC)[15], with 1020 groups reported in 2018 after the number had declined to 784 in 2014, and concerns over the rising prevalence of online hate speech have led to shifts in social media policies[16–18].

These trends highlight the importance of developing a better understanding of acts of hate. Research addressing acts of hate has often focused on the role of inter-group threat as a focal mechanism in the emergence of behaviors, such as hate crime[4,19], hate group activity[20–23], and hate speech[24]. Echoing these findings, psychological investigations of prejudice have observed that perceptions of either realistic or symbolic outgroup threat[25,26] lead to increased prejudice toward outgroups and that this effect is mediated by attitudes associated with right-wing authoritarianism and social dominance[27–31].

Together, this line of work suggests behaviors like hate crime, hate group activity, and hate speech can be at least partly understood as responses to perceived outgroup threats[4]. However, this account raises an essential question: what is it about some perceived threats—and the people who perceive them—that contributes to acts of hate?

To answer this question, we propose that the moralization of a perceived threat is a central factor in the process underlying acts of hate, such as hate speech and hate group activity—behaviors we refer to collectively as extreme behavioral expressions of prejudice (EBEPs). This view is grounded in a large body of research linking violence and extreme behavior to moral values, perceptions of moral violations, and feelings of moral obligation[32–42]. Drawing on this work, we suggest EBEPs are often motivated by the belief that an outgroup has done something morally wrong and, further, that a person's risk of perceiving such moral violations is partially dependent on their own moral values—a hypothesis we refer to as the moralized threat hypothesis.

This hypothesis is specifically informed by Fiske and Rai[32], who find many forms of violence are morally motivated, such that people often feel that hurting others is fundamentally right. They find this is the case even for the kinds of violent behaviors most people regard as morally repugnant. In this sense, Fiske and Rai[32] argue that most violence is morally motivated.

From a broader social perspective, morally motivated violence can also be understood as moral backfiring. That is, adaptive moral values that promote social cohesion in many contexts, such as moral values oriented around loyalty and respect, can serve as the foundation for morally repugnant behavior. Accordingly, we hypothesize acts of hate are often motivated by the belief that the outgroup that has been targeted by abusers has violated a moral value[43].

Given recent increases in EBEPs aligned with right-wing ideology[10,12,44] and concerns over the role of hate speech in violent crimes toward social identities often demonized by right-wing extremist groups[45], we focused largely on EBEPs that were aligned with right-wing ideologies. Accordingly, we expected that these EBEPs would be associated with moral values oriented around group preservation because such values have been linked to conservatism and right-wing ideologies in US contexts[46,47].

To operationalize these values, we rely on moral foundations theory (MFT)[48,49], which proposes a hierarchical model of moral values composed of two superordinate, bipolar categories: individualizing values and binding values. While the former is comprised of values focused on individuals' rights and well-being—caring for others and following principles of fairness—the latter is comprised of values considered to be associated with group preservation—maintaining ingroup solidarity, submitting to authority, and preserving the purity of the body and sacred objects. The individualizing foundations (care/harm and fairness/cheating) are theorized to sensitize people to human suffering and equitable outcomes in the society. Binding foundations (loyalty/betrayal, authority/subversion, and purity/degradation) on the other hand are theorized to sensitize people to the group's well-being and maintenance of hierarchy[48]. MFT has been repeatedly applied to study liberal-conservative differences, finding that liberals endorse the individualizing values more than the binding values, whereas conservatives endorse all five foundations more or less equally[46,47].

Using this model of moral values, the moralized threat hypothesis predicts that binding values are associated with EBEPs toward groups marginalized by the ideological right. We examine this prediction across five studies. In Study 1, we focus on the geospatial relationship between US county-level moral values and the county-level prevalence of hate groups. Then, in Studies 2–5, we switch to data collected via surveys, which enable us to test our hypotheses with more precision and control. In these studies, we investigate whether people see a range of EBEPs as more justified when they believe that the targeted outgroup has done something morally wrong. Our theoretical process model is shown in Fig. 1.

## Results

**Study 1: Hate groups and county-level moral values in the US.** In Study 1, we test the viability of the moralized threat hypothesis by considering the relationship between hate group activity and moral values. Specifically, we test the hypothesis that county-level moral values—i.e., the individualizing and binding values—are associated with the county-level prevalence of hate groups in the US obtained from the SPLC. Although SPLC conducts extensive data collection efforts, there is no conclusive evidence that the data are complete or without bias. Recent research has found positive associations between regional variations in Uniform Crime Reporting (compiled by the FBI) and the geographic distribution SPLC-identified hate group activities[50]. Finally, because the data used to estimate county-level moral values were collected from 2012 to 2018, we calculated the average county-level count

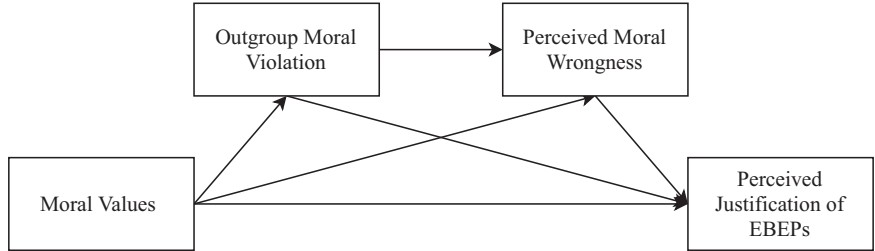

**Fig. 1 Moralized Threat Hypothesis.** The theoretical process model for the moralized threat hypothesis.

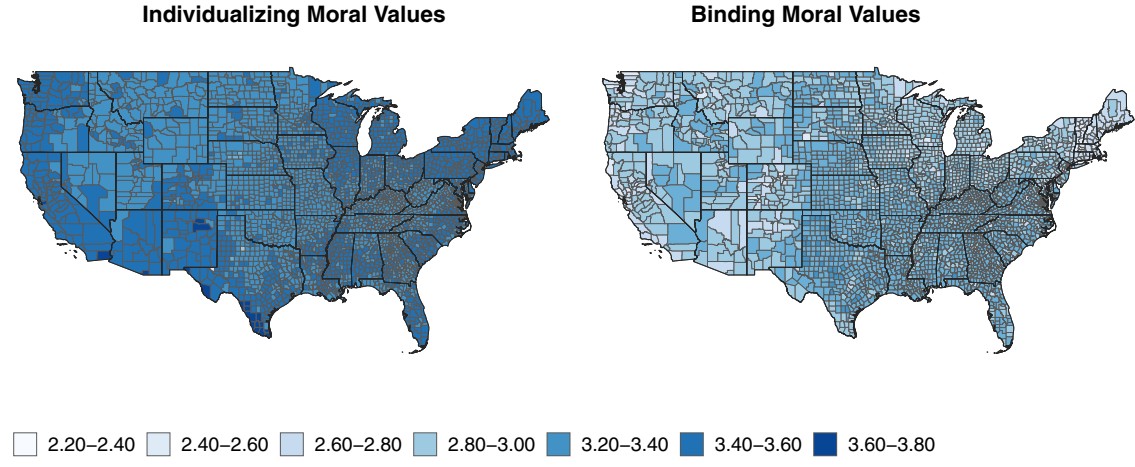

**Fig. 2 County-level binding and individualizing values.** Estimates of county-level individualizing and binding moral values adjusted for representativeness via MrsP with spatial smoothing.

of active hate groups from 2012 to 2017 (the latest available year in the SPLC data).

The distributions of county-level binding and individualizing values (See Fig. 2) were estimated from data collected via YourMorals.org using Multilevel Regression and Synthetic Poststratification (MrsP)[51], a model-based approach to survey adjustment and sub-national estimation that extends Multilevel Regression and Poststratification (MrP)[52]. Finally, we used these county-level estimates of individualizing and binding values to predict the county-level prevalence of hate groups.

Comparisons of model predictions of the county-level rate of hate groups were largely consistent with the observed rates (See Fig. 3), root mean squared error (RMSE) = 0.15. Consistent with our hypotheses, our results indicate a relationship between the county-level rate of hate groups and county-level binding values (See Fig. 4). Even after adjusting for county-level ethnic composition, educational attainment, the proportion of county population below the poverty line, 2016 Democratic Presidential vote share, and being rural vs. urban, an additional standard deviation in binding values is associated with an odds ratio of 1.32 (posterior SD = 0.14, 95% HPDI = [1.05, 1.61]) for the rate of hate groups. Notably, no such effect was observed for individualizing values.

As expected, when state-fixed effects were added to the model, the association between binding values and hate group rates attenuated toward zero by about 40% and its standard error increased by roughly 80%. As such, under the fixed effects estimator, the association between binding values and the rate of hate groups is not distinguishable from null. This is notable, because it suggests that unmeasured state-level confounds could be driving the association observed in the model that excluded state-level fixed effects. That said, these results can also be

explained simply by the fact that fixed effects estimators are likely to be underpowered when applied to data with low within-group variation, such as the current study.

**Study 2: Perceived justification of EBEPs against muslims in the US.** In Study 1, our results indicated an association between EBEPs and moral values in the context of the spatial distribution of hate groups in the US. In the next series of studies, we extend this result by investigating the relationship between people's moral values and the degree to which they believe EBEPs are justified. Specifically, we focus on four distinct EBEPs: posting hate speech on Facebook, sharing hate speech on flyers, verbally assaulting an outgroup member, and physically assaulting an outgroup member (See Supplementary Information for the EBEP items used). Specifically, we test three primary hypotheses from the moralized threat hypothesis:

(1)  Hypothesis 1. An EBEP toward an outgroup is perceived as more justified when that outgroup is perceived as having done something morally wrong.
(2)  Hypothesis 2. EBEPs should be perceived as more justified by people who prioritize binding values.
(3)  Hypothesis 3. The association between binding values and justification of EBEPs toward a given outgroup is at least partially mediated by the perception that the outgroup has done something morally wrong.

First, we investigate the relationship between binding values and the perceived justification of EBEPs against an outgroup in the US, Muslims. Specifically, we focus on the perceived moral wrongness of Muslims allegedly "spreading Islamic values" in the US. We chose this social group due to the cultural salience of Muslims as an outgroup, recent increases in hate crimes against

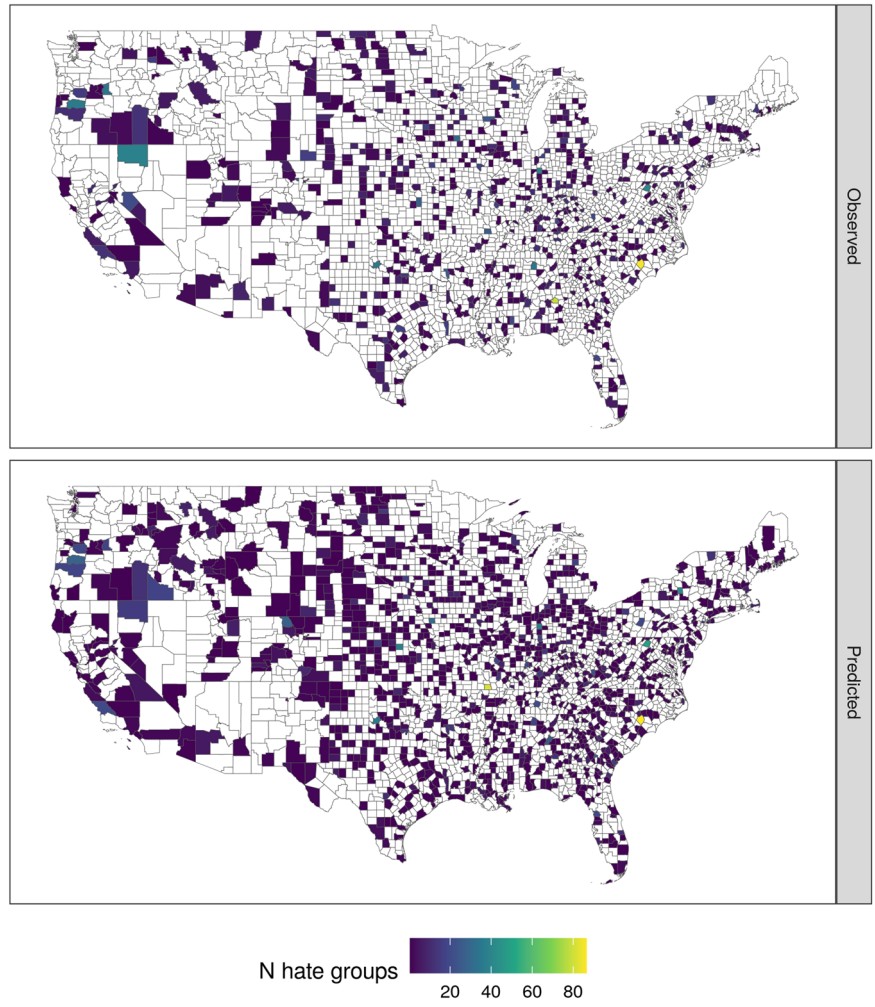

**Fig. 3 Observed (Top) and predicted (Bottom) county-level count of hate groups.** Counties with observed or estimated values of zero are set to NA to provide contrast with positive observations.

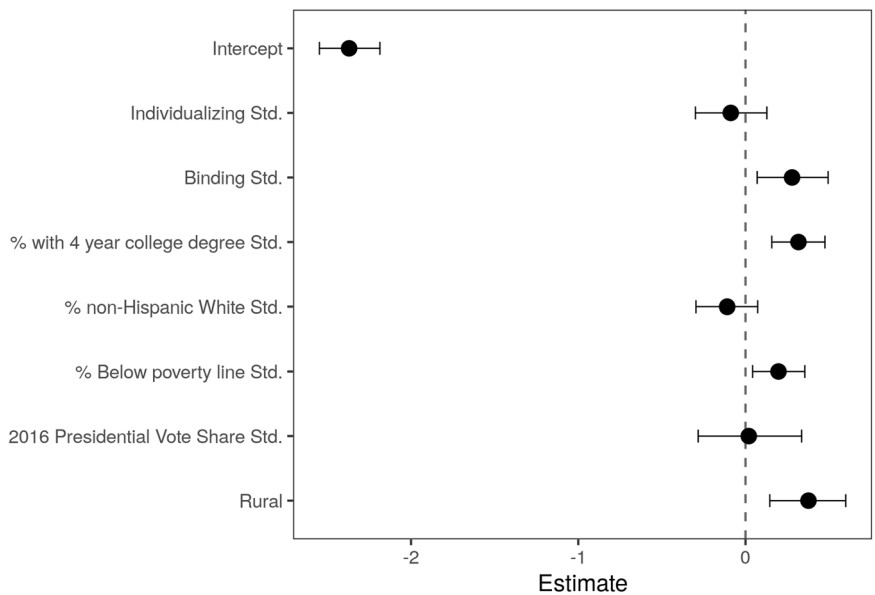

**Fig. 4 Estimated conditional association between binding values and the county-level prevalence of hate groups.** Error bars represent 95% HDI.

Muslims[53], and because focusing on "spreading Islamic values" enabled us to evaluate the moralized threat hypothesis under conditions of symbolic threat (i.e., the symbolic threat of divergent values).

To test our hypotheses, we collected responses to a survey designed to measure participants' binding and individualizing values, the degree to which they believed it is immoral for Muslims to be "spreading Islamic values" in the US, and the degree to which they felt that EBEPs against Muslims were justified. Using these data, we estimated a series of Bayesian regression models. First, we modeled (Model 1) the association between participants' standardized individualizing and binding values and the degree to which they perceived EBEPs against Muslims as justified. In this model, we treated responses to the EBEP items (Cronbach's $\alpha = 0.92$, 95% CI = [0.90, 0.93]) as a repeated measure of EBEP justification, yielding four measurements for each participant. To account for this, we used a Bayesian hierarchical modeling framework to allow for varying intercepts (i.e., random effects) for both participants and EBEP items. This approach enabled our model to address the facts that (1) different participants should be more or less likely to see EBEPs as justified in general and (2) that each EBEP item, on average, should be seen as more or less justified. Further, to account for the fact that the effects of experimental condition on EBEP justification may vary depending on the EBEP, we also allowed the effect of condition to vary across EBEP items (i.e., by estimating random slopes). Finally, because the distribution of responses to each EBEP item was heavily skewed, we modeled EBEP justification using a cumulative logistic regression model[54]. Altogether, this yielded a hierarchical Bayesian cumulative logistic regression model in which (1) EBEP justification was regressed on experimental condition and (2) varying intercepts for participant and EBEP item and a varying slope for both conditions were estimated.

To test hypotheses 2 and 3, we estimated two additional regression models, one (Model 2) in which the perceived moral wrongness of Muslims "spreading Islamic values" was regressed on participants' individualizing and binding values and another (Model 3) in which the perceived justification of EBEPs against Muslims was regressed on participants' individualizing and binding values as well as perceived moral wrongness. Finally, we used Bayesian posterior simulations[55–58] to estimate the degree to which perceived moral wrongness statistically mediated the effect of binding values on the perceived justification of EBEPs against Muslims.

Consistent with our hypotheses, estimates from model 1 indicated a strong association between participants' binding values and the degree to which they perceived EBEPs against Muslims to be justified, $b = 2.29$, posterior SD = 0.35, 95% CI = [1.67, 2.96] (See Fig. 5). We also observed a negative effect of individualizing values, $b = -1.70$, posterior SD = 0.32, 95% CI = [-2.32, -1.10]. Further, estimates from model 2 showed a positive association between perceived moral wrongdoing (PMW) and EBEP justification, $b = 1.72$, posterior SD = 0.51, 95% CI = [0.75, 2.77]. Adjusting for PMW, in model 2, also led to a substantial reduction, relative to model 1, in the magnitude of the effects of individualizing values, $b = -1.29$, posterior SD = 0.34, 95% CI = [-1.89, -0.68], and binding values, $b = 1.48$, posterior SD = 0.35, 95% CI = [0.82, 2.18]. Finally, our mediation analysis indicated that perceived moral wrongness statistically mediated the association between binding values and perceived justification of EBEPs (See Supplementary Information for full average mediation effects). Notably, we also conducted a secondary mediation analysis in which participants' political ideology (measured on a 7-point "Very Liberal" to "Very Conservative" scale) was included as an adjustment variable. This revealed that adjusting for political ideology did not lead to substantive changes in any of our results (See Supplementary Information Study 2 for details).

**Study 3: Perceived justification of EBEPs against Mexican immigrants in the US**. In this study, we sought to conceptually replicate the effects observed in Study 2 with a different social outgroup in the US, Mexican immigrants. We focus on Mexican immigrants in the US due to their cultural salience as a social group and consistent increases hate crimes targeting Mexicans living in the US in recent years[59]. To test our hypotheses, we rely on the same analytical procedure used in Study 2.

Consistent with Study 1, model estimates in the present study indicated a strong association between participants' binding values and the degree to which they believed EBEPs against Mexican immigrants in a fictional town in the US, Webster Springs, were justified (See Fig. 5; See Supplementary Information for the vignette). Specifically, after attempting to account for the effect of individualizing values, the odds of selecting a higher, vs. lower, response option were estimated to be 4.95 times greater given a standard deviation increase in binding values, $b = 1.60$, posterior SD = 0.27, 95% CI = [1.12, 2.14]. In contrast, this model indicated that individualizing values were negatively associated with EBEPs, such that the odds of selecting a higher, vs. lower, response option were estimated to be 0.31 times lower given a standard deviation increase in individualizing values, $b = -1.15$, posterior SD = 0.41, 95% CI = [-1.88, -0.35].

Next, as in Study 2, we tested hypotheses 1 and 3 via two additional regression models, one (Model 2) in which the perceived moral wrongness of Mexican immigrants allegedly taking jobs was regressed on participants' individualizing and binding values and another (Model 3) in which the perceived justification of EBEPs against Mexicans was regressed on participants' individualizing and binding values as well as perceived moral wrongness. As expected, estimates from model 2 indicated a positive association between participants' binding values and the degree to which believed it was morally wrong for Mexican immigrants to "take jobs" in Webster Springs, $b = 0.47$, posterior SD = 0.05, 95% CI = [0.37, 0.56]. That is, a standard deviation increase in binding values was associated with an estimated 0.47 standard deviation increase in perceived moral wrongness. In contrast, individualizing values were estimated to be negatively associated with the perception of moral wrongdoing, $b = 0.47$, posterior SD = 0.05, 95% CI = [0.37, 0.56].

Further, as hypothesized, estimates from model 3 indicated that even after attempting to adjust for the effects of standardized individualizing and binding values, standardized PMW was estimated to be positively associated with perceived EBEP justification, $b = 1.63$, posterior SD = 0.46, 95% CI = [0.70, 2.49]. Thus, the odds of seeing EBEPs as more justified than a given response level vs. less justified or equal to that level are 5.10 times higher given a standard deviation increase in PMW. Notably, adjusting for the effect of PMW also substantially decreased the estimated effects of binding ($b = 0.73$, posterior SD = 0.26, 95% CI = [0.24, 1.23]) and individualizing ($b = -0.87$, posterior SD = 0.46, 95% CI = [-1.65, 0.04]) values.

Finally, similar to Study 2, we relied on posterior simulation to test the hypothesis that PMW statistically mediates the association between binding values and EBEP justification. Results from this analysis indicated that PMW partially mediated the association between binding values and perceived EBEP justification (See Supplementary Information). Similar to Study 2, we found that including political ideology in these analyses did not lead to substantive changes in any of our results.

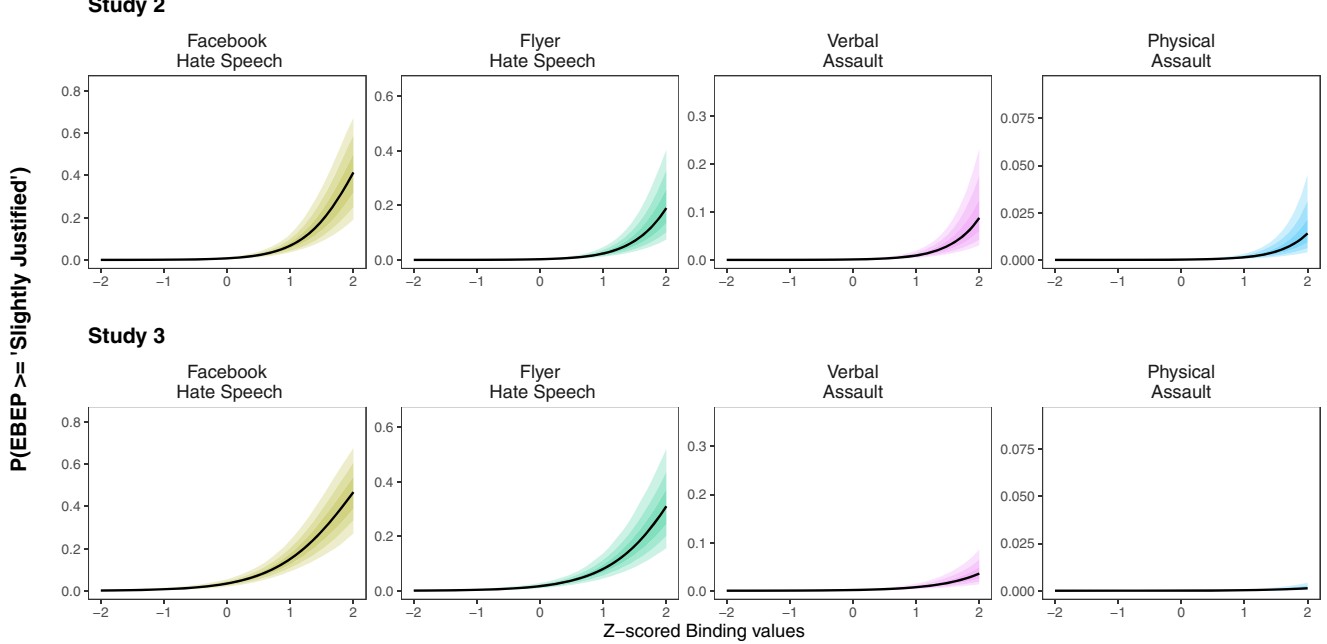

**Fig. 5 Estimated association between binding values and perceived justification of EBEPs in Study 2 ($N = 511$) and Study 3 ($N = 355$).** Participants who prioritized binding values tended, on average, to perceive EBEPs against outgroups as more justified. Shaded bands represent predicted values from simulations of the respective model.

**Study 4: Experimental manipulation of perceived moral wrongness.** While Studies 2 and 3 are consistent with the moralized threat hypothesis, they both rely on observational designs and thus cannot directly address the question of whether perceptions of outgroup moral wrongdoing cause increases in the perceived justification of EBEPs against the outgroup. We address this issue by directly manipulating participants' perceptions of a fictional social outgroup, 'Sandirians' (See Supplementary Information for the vignettes used). To do so, we randomly assigned participants to one of two conditions—a 'high moral threat' condition and a 'low moral threat' condition—that manipulated the moral valence of a fictional outgroup. We focus on a fictional outgroup in order to limit the influence of participants' prior beliefs on their responses[60]. As such, the goals of this study were to estimate the causal effect of an outgroup's ostensibly immoral behavior—the construct that we experimentally manipulate—on the degree to which EBEPs against the outgroup were seen as justified and, further, to investigate whether PMW mediates this causal effect.

To test the hypothesis that PMW mediates the effect of outgroup moral behavior on EBEP justification, we first estimated a Bayesian linear regression (Model 1) in which z-scored PMW was regressed on experimental condition. Estimates from this model (See Table 15 in Supplementary Information for complete model estimates) show that participants in the high perceived moral threat condition reported substantially higher levels of PMW, $b = 0.50$, posterior SD $= 0.11$, 95% CI $= [0.28, 0.73]$.

Next, we assessed the effect of experimental condition on perceived justification of EBEPs. To do this, we estimated a second model (Model 2) in which perceived justification of EBEPs was regressed on experimental condition. In this model, we again treated responses to the EBEP items (Cronbach's $\alpha = 0.84$, 95% CI $= [0.82, 0.86]$) as a repeated measure and we used the same hierarchical Bayesian ordered logistic regression model used in the previous studies.

Results from model 2 (See Table 15 for complete model estimates) indicate that, even after attempting to account for the random effects of participant (intercept SD $= 3.79$) and EBEP

item (intercept SD $= 4.04$, $b_{threat}$SD $= 0.62$), participants in the high perceived moral threat condition were substantially more likely to see EBEPs against Sandirians as more justified, compared with participants in the low perceived moral threat condition, $b = 1.44$, posterior SD $= 0.68$, 95% CI $= [0.16, 2.80]$, OR $= 4.21$, 95% CI $= [1.18, 16.45]$. In other words, for participants in the high perceived moral threat condition, the odds of seeing EBEPs, on average, as extremely justified, vs. less than extremely morally justified, was 4.21 times higher than for participants in the low perceived moral threat condition.

Next, we directly investigated the role of PMW in participants' EBEP justification responses. To do this, we extended model 2 by including standardized PMW—the degree to which participants believed it was morally wrong for Sandirians to take jobs in Webster Springs—as an independent variable with varying slopes across EBEP items (Model 3). Estimates from this model indicated a strong positive association between believing it was morally wrong for Sandirians to take jobs and perceiving EBEPs against Sandirians as more justified, $b = 2.21$, posterior SD $= 0.42$, 95% CI $= [1.48, 2.94]$, OR $= 9.20$, 95% CI $= [4.38, 18.92]$. Notably, adjusting for PMW also led to a dramatic attenuation of the estimated effect of experimental condition, such that a clear positive effect was no longer supported, $b = 0.28$, posterior SD $= 0.63$, 95% CI $= [-0.86, 1.57]$, OR $= 1.29$, 95% CI $= [0.42, 4.79]$.

Finally, we tested the hypothesis that PMW mediated the effect of experimental condition on EBEP justification. Using Bayesian posterior simulation, we found that PMW statistically mediates the effect of experimental condition on the probability of indicating that an EBEP was at least slightly justified (See Supplementary Information).

Consistent with hypothesis 1, these results indicate that participants who were led to believe that a fictional immigrant group—the Sandirians—had done something immoral also believed that EBEPs against this group were more justified, compared with participants in the control condition. Importantly, adjusting for the degree to which participants believed Sandirians had done something morally wrong also completely accounted for the effect of experimental condition. Consistent with this,

mediation analyses also indicated that the effect of experimental manipulation was mediated by the degree to which participants believed that the Sandirians had done something immoral. Importantly, these effects hold even after adjusting for the degree to which participants identify as conservative.

**Study 5: Differential effects in the domain of binding values.** In the previous studies, we found observational and experimental evidence that people perceive EBEPs against outgroups as more justified when they believe the outgroup has done something considered to be morally wrong. However, the moral violations that those studies focused on were potentially confounded with political ideology and could also be understood simply as existential (Study 2) or economic threats (Studies 3 and 4). Further, none of the previous studies address the question of whether the implications of the moralized threat hypothesis operate equivalently outside the domain of binding values.

In Study 5, we address these issues by experimentally manipulating perceived moral threats that do not pose any clear or direct existential or economic threats (See vignettes in Supplementary Information; please note the study materials containes graphic content.). Further, we focus on both binding and individualizing moral threats that are not clearly confounded with ideology. Under this approach, we are able to evaluate the differential effects of binding and individualizing values violations on EBEP justification. Specifically, we again test the hypothesis that PMW mediates the effect of the experimental manipulation of perceived moral threat on the perceived justification of acts of hate. However, in this study, we also investigate whether this mediation effect holds for both binding and individualizing values violations. We also directly investigate the role of binding and individualizing values in the causal chain implied by the moralized threat hypothesis by testing for whether people's binding and individualizing values moderate the mediation effect of PMW. We conduct these tests of moderated mediation because, conditional on the moralized threat hypothesis, exposing participants to a violation of binding values (vs. individualizing values) should only increase the perceived justification of hate acts for people who prioritize the binding values (vs. individualizing values). As such, in this study, we conceptualize binding values (vs. individualizing values) as a treatment susceptibility factor that amplifies or dampens the chain connecting the treatment, PMW, and the perceived justification of hate acts.

These tests of moderated mediation allow us to address four alternative hypotheses:

5.1 Neither binding nor individualizing values function as treatment susceptibility factors

5.2 Both binding and individualizing values function as a treatment susceptibility factor

5.3 Only individualizing values function as a treatment susceptibility factor

5.4 Only binding values function as a treatment susceptibility factor

Evidence for hypothesis 5.1 would suggest that our perceived moral threat manipulations were not meaningfully targeting participants' binding and individualizing values. This would constitute evidence against the moralized threat hypothesis. Alternatively, evidence for hypothesis 5.2 would suggest that our perceived moral threat manipulations do meaningfully target participants' binding and individualizing values, but also that there is nothing unique about the effect of binding values and binding values violations. Finally, evidence for hypotheses 5.3 or 5.4 would indicate that the domain of individualizing values or binding values, respectively, have a unique effect on EBEP

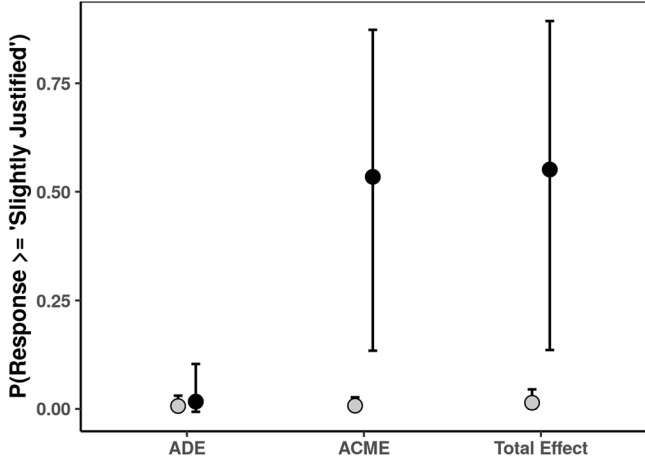

**Fig. 6 Estimated moderated average causal mediation effect for binding values violations at low (−1 SD) and high (+1 SD) binding values ($N_{control} = 349$, $N_{binding} = 348$).** Error bars represent 95% CI.

justification. As such, this design enables us to directly address the questions of whether the implications of the Moralized Threat Hypothesis are equivalent in the domains of binding and individualizing values.

Prior to testing our moderated mediation hypotheses, we first confirm that our manipulations of binding and individualizing values violations successfully targeted the relevant foundation. As expected, when exposed to binding values violations, participants who prioritized the binding values reported higher PMW, $b = 0.30$, posterior SD $= 0.07$, 95% CI $= [0.16, 0.44]$; however, no such effect was observed for participants who prioritized individualizing values, $b = 0.07$, posterior SD $= 0.07$, 95% CI $= [−0.21, 0.07]$. In contrast, for participants exposed to individualizing values violations, this pattern reversed. Participants who prioritized individualizing values reported higher levels of PMW, $b = 0.30$, posterior SD $= 0.07$, 95% CI $= [0.17, 0.43]$; however, no such effect was observed for participants who prioritized binding values $b = −0.09$, posterior SD $= 0.07$, 95% CI $= [−0.22, −0.04]$.

To test for moderated mediation, we first investigate whether binding values moderate the mediation effects of binding and individualizing values violations. Specifically, to test whether binding values moderate the mediating effect of perceived moral wrongness, we estimate the Average Causal Mediation Effects (ACME) for perceived moral wrongness with binding values fixed at high (1 SD above the mean) and low (1 SD below the mean) values[61]. Under this design, evidence of moderated mediation is constituted by an increase in the ACME when binding values are high compared with the ACME when binding values are low. As such, we treat binding values as a treatment susceptibility factor that influences the degree to which participants respond to the experimental manipulation of binding values.

Using an analytic strategy similar to those employed in the previous studies, we estimated two Bayesian regression models in order to test for moderated mediation in the domain of binding values violations. Our results were strongly consistent with the hypothesis of moderated mediation (See Fig. 6). For simplicity, for this analysis and the following analyses, we report the moderated mediation effects for the probability of indicating acts of hate are at least slightly justified, marginalized across the type of hate act (See Supplementary Information for tables reporting full moderated mediation effects). Specifically, there was no

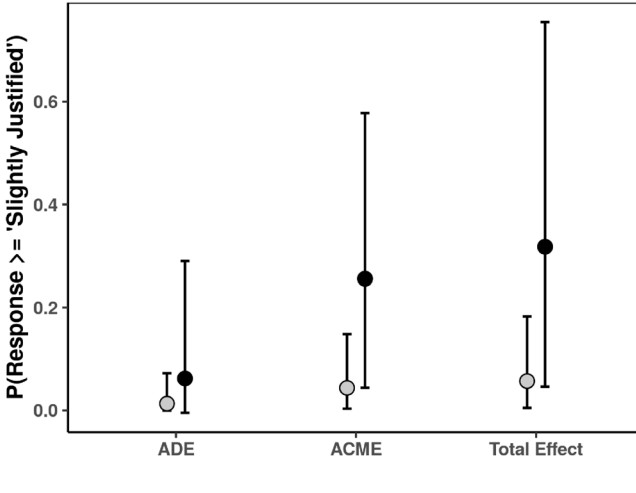

**Fig. 7 Estimated moderated average causal mediation effect for individualizing values violations at low (−1 SD) and high (+1 SD) binding values ($N_{control}$ = 349, $N_{individualizing}$ = 349).** Error bars represent 95% CI.

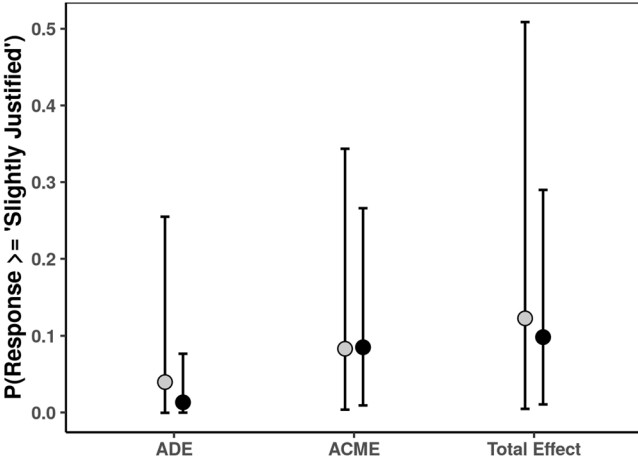

**Fig. 8 Estimated moderated average causal mediation effect for individualizing values violations at low (−1 SD) and high (+1 SD) individualizing values ($N_{control}$ = 349, $N_{individualizing}$ = 349).** Error bars represent 95% CI.

evidence for a substantive Average Direct Effect (ADE) of experimental manipulation for participants with low binding values ($ADE_{low}$) or high binding values ($ADE_{high}$), $ADE_{low}$ = 0.01, posterior SD = 0.02, 95% CI = [−0.0003, 0.031], $ADE_{high}$ = 0.02, Posterior SD = 0.04, 95% CI = [−0.007, 0.10]. Further, for participants with low binding values, there was also no evidence for a substantive ACME, $ACME_{low}$ = 0.01, posterior SD = 0.01, 95% CI = [0.0007, 0.026]. However, for participants with high binding values, there was strong evidence for a large ACME, $ACME_{high}$, 0.53, posterior SD = 0.21, 95% CI = [0.13, 0.87].

Next, we conducted an analogous test in the domain of individualizing values violations (See Supplementary Table 24 for model estimates). Per our treatment susceptibility hypotheses, the moderation effect of binding values in the domain of individualizing violations should be smaller than in the domain of binding

values. Under a perfect experimental individualizing values violation, we would expect this moderation effect to be indistinguishable from null. However, given that the individualizing values violation may also induce some degree of binding violations (i.e., it is likely not a perfect experimental manipulation of individualizing violation), a small moderation effect would not be inconsistent with our hypotheses. Consistent with these expectations, we found that the estimated ACME under high binding values ($ACME_{high}$ = 0.25, posterior SD = 0.15, 95% CI = [0.04, 0.57]) was less than half of the analogous effect in the binding values violation domain. Further, while our point estimate of the ACME under low binding values ($ACME_{low}$ = 0.05, posterior SD = 0.06, 95% CI = [0.004, 0.16]) is smaller than the point estimate of $ACME_{high}$, the CIs of these parameters overlap substantively (See Fig. 7).

Finally, we used the same analytic procedure to investigate whether participants' individualizing values moderate the ACME of the individualizing values violation condition. Our results suggested that in the domain of individualizing values violations, the ACME of perceived moral wrongness is not moderated by individualizing values, $ACME_{low}$ = 0.08, posterior SD = 0.10, 95% CI = [0.004, 0.34] and $ACME_{high}$, 0.09, posterior SD = 0.07, 95% CI = [0.009, 0.27]. That is, we found no evidence that individualizing values function as a treatment susceptibility factor (See Fig. 8). Accordingly, our results support hypothesis 5.4: only binding values serve as a treatment susceptibility factor and, importantly, this effect is the most potent in the domain of binding values violations (See Supplementary Information for additional analyses and results).

## Discussion

Taken together, our geospatial analysis of 3108 US counties (Study 1), and social psychological experiments on over 2200 participants (Studies 2–5) converge on the finding that perceived justification of EBEPs is tied to people's group-oriented moral values. In Study 1, county-level moral values, particularly binding values, were found to predict hate group activity in US counties. Specifically, while both individualizing and binding values were predictive of hate group activity, only binding values remained predictive after controlling for county-level covariates, such as political ideology. Even though we argue that local spatial smoothing is the more appropriate analysis for our data because (1) neither binding values nor hate groups are meaningfully constrained by state boundaries and (2) fixed effects estimators are likely to be underpowered when applied to data with low within-group variation, we note that, when state-fixed effects were added to the model, the relationship between binding values and the rate of hate groups was not distinguishable from null. Future research should further investigate the discrepancies in estimates achieved using local spatial smoothing compared to those achieved with states as fixed effects.

To further dissect the mechanisms responsible for the association between binding values and the perceived justification of EBEPs, we conducted a series of controlled experiments (see Fig. 1 for our process model). These results suggest that the association between binding values and justifying acts of hate depends, at least partially, on people's perceptions of outgroup moral violations. In Studies 2 and 3, we found that this association was positively mediated by PMW for two marginalized groups in the U.S. context, Muslims (Study 2) and Mexican immigrants (Study 3), even after adjusting for participants' political ideology. Further, in Studies 4 and 5, we experimentally replicated the effect of PMW on the justification of acts of hate toward two different fictional outgroups.

Notably, Study 5 provided tentative evidence that binding values may be a particularly important risk factor for the perceived justification of EBEPs. Participants who were experimentally manipulated to believe an outgroup had done something immoral were more likely to perceive acts of hate against that outgroup as justified when they felt that the outgroup's behavior was more morally wrong. However, this association between PMW and the justification of hate acts was strongly moderated by people's binding values, but not by their individualizing values. Ultimately, comparing people high on binding values to people high on individualizing values, we found that the average causal mediation effect in the domain of binding values was more than six times the average causal mediation effect in the domain of individualizing values. In other words, our results suggest that if two people see an outgroup's binding values violation as equally morally wrong, but one of them has higher binding values, the person with higher binding values will see EBEPs against the outgroup as more justified. However, no such difference was observed in the domain of individualizing values.

Accordingly, our results suggest that people who attribute moral violations to an outgroup may be at higher risk for justifying, or perhaps even expressing, extreme prejudice toward outgroups; however, our results also suggest that people who prioritize the binding values may be particularly susceptible to this dynamic when they perceive a violation of ingroup loyalty, respect for authority, and physical or spiritual purity. In this sense, our findings are consistent with the hypothesis that acts of hate—a class of behaviors of which many have received their own special legal designation as particularly heinous crimes[4]—are partly motivated by individuals' moral beliefs. This view is well-grounded in current understandings of the relationship between morality and acts of extremism or violence[33,35–37,39,62,63].

Given today's digital media environment and its potential for stoking moral outrage[64] and uniting isolated individuals who share fringe ideologies, understanding these effects is particularly important[65]. While much research on EBEPs has highlighted the role of specific, concrete threats[24,66], the moralized threat hypothesis offers a potential framework for understanding why people may perceive EBEPs as justified, even in the absence of an ostensible material threat. This hypothesis suggests that a person does not necessarily need to fear for their job or safety to engage in or approve of EBEPs; instead, it may be sufficient for them to simply feel a sense of moral outrage[67]. That said, our work tentatively suggests that different domains of moral values may have different risk functions with regard to acts of hate.

Our findings tentatively suggest that binding values are a more severe risk factor for acts of hate, relative to individualizing values. Further, research suggests that, at least in the United States, binding values are held more strongly among people who report right-wing political ideology[46,47]. Accordingly, our work suggests that people who strongly endorse right-wing ideologies in the US may be at higher risk for engaging in the justification of EBEPs. Notably, while we tried to test our model in real-world and fictional outgroups with high and low threat levels, it is possible that the generality of our model is constrained to groups who specifically coalesce around loyalty, authority, and purity. Hence, future research is encouraged to examine the perceived justification of EBEPs in cases where groups coalesce around fairness and harm avoidance.

Ultimately, in the current work, we advance the current understanding of these dynamics by providing a diverse set of evidence that suggests that moral values and the perception of moral wrongdoing play an important role in EBEPs. It may be that further accounting for these factors can help improve our understanding of when and why feelings of threat or prejudice do not just manifest in negative social attitudes or low-risk expressions of prejudice, but rather erupt into EBEPs.

## Methods
All our studies were approved by University of Southern California's Institutional Review Board (IRB) under UP-19-00395. In Studies 2, 3, 4, and 5, before participating in the studies, all participants were provided an information sheet, approved by USC's IRB, explaining the studies. At the end of the experiments, all participants were debriefed, thanked, and provided with a text presenting the purpose of the study (see Supplementary Information).

### Study 1
*Estimating county-level moral values.* To estimate the county-level distribution of moral values, we use data collected via YourMorals.org, a website operated by the founders of MFT to collect measurements of voluntary respondents' moral values, from ~2012 to 2018 ($N = 106,465$, 16% conservative, 13% moderate 71% liberal, 47% women, see Table 1 in Supplementary Information for joint distribution of Age and Education sample proportions). While this is a relatively large sample, it cannot be used to directly estimate county-level moral values because it is not randomly sampled or representative at the county level[68]. To account for these issues, we rely on MrsP[51], a model-based approach to survey adjustment and subnational estimation that extends MrP[52].

Both MrP and MrsP involve estimating regional outcomes on a target construct from individual-level data. These data are used to model the target construct as a multilevel function of demographic characteristics (e.g., gender, age, and level of education), regional indicators (e.g., county, state, or region), and regional factors (e.g., presidential vote proportion, median income, or educational attainment). This model is then used to generate predictions for each combination of demographic characteristics within each region. Finally, information about the population distribution of these demographic characteristics within each region are used to estimate a weighted mean based on the model predictions.

Here, we use adjusted MrsP, which follows the above approach, but also enables the inclusion of a more diverse set of demographic variables. Adjusted MrsP accomplishes this by incorporating additional information about the joint distribution of demographic variables derived from an external source. In our case, we rely on the Pew Foundation's Religious Landscape Study data[69], a nationally representative data consisting of a sample of over 37,000 Americans. Specifically, we model individual-level responses to each moral foundation as a function of six demographic variables: gender (2 levels), age (3 levels), ethnicity (4 levels), level of education (3 levels), religious attendance (3 levels), and political ideology (3 levels). We also account for two levels of regional clustering, the county level and the region level and include the proportion of Democratic votes in the 2016 presidential election as a county-level factor. Finally, the multilevel model that we estimate also includes a hierarchical auto-regressive prior[70] that, under the presence of spatial autocorrelation, induces local spatial smoothing[68,71,72] between proximate counties. (For a detailed discussion of these methodologies and models, as well as an evaluation of the efficacy of using this approach with non-random, non-representative data, see ref. [68]).

In order to evaluate the degree to which MrsP could be used to derive reliable county-level estimates from the YourMorals data, we first estimated the county-level distribution of conservatives, an outcome for which we could obtain a meaningful gold-standard measurement to validate against[68]. Specifically, we estimated the county-level proportion of self-identified conservatives using the same MrsP procedure that we use to estimate county-level moral values. However, one key difference is that we did not include any context-level measures of political ideology or vote share in the MrsP response model. More specifically, at the individual level, the response model adjusted for respondent ethnicity, gender, level of education, age, and frequency of religious attendance (these variables were identical to those we include in the response models used to estimate county-level moral values). At the context level, we included the county-level rate of protestant evangelicals, which we obtained from the 2010 Religious Census[73]. Then, to investigate the degree to which MrsP can correct for self-selection bias in the YourMorals data, we evaluated the association between county-level 2012 and 2016 Presidential election Republican vote shares and our estimates of county-level conservatism.

Results from our validation analyses indicated that MrsP was able to sufficiently account for response biases in the YourMorals data. Specifically, our county-level estimates of conservative residents was a reasonable approximation of the 2016 Republican Presidential vote share, $r (3104) = 0.63$, $t = 44.81$, $p < 0.001$, 95% CI = [0.61, 0.65], RMSE = 0.20 (See Figs. 1 and 2 in Supplementary Information for scatter plot and geographic visualization, respectively).

Next, we estimated the county-level distribution of each moral foundation (an interactive visualization of these estimates can be viewed at https://mapyourmorals.usc.edu/) and then used these estimates to calculate scores for the individualizing (Care and Fairness) and binding (Loyalty, authority, and purity) dimensions for each county.

*County-level distribution of hate groups.* Estimates of the county-level distribution of hate groups were obtained from the SPLC[74], which maintains an ongoing hate

group task force that monitors and documents hate group activity at the city level. We used these data to generate county-level counts of active hate groups by identifying the counties containing each city. For cities located in multiple counties, we selected the county containing the largest proportion of the city's population in order to avoid over counting hate groups. Of note, the SPLC data are not without limitations. Hate group activities are collected and indexed in their database by volunteers and staffers, hence they may be subject to organizational and personal biases[75].

*Estimating the association between county-level rate of hate groups and moral values.* Using the SPLC measurement of hate-group prevalence as the outcome, we estimated the county-level rate of hate groups per 10,000 inhabitants using a negative-binomial regression model with Bayesian Spatial Filtering[76] to account for residual spatial autocorrelation without inducing spatial confounding with the independent variables included in our model[77]. As predictors in this model, we included standardized estimates of individualizing and binding values as the independent variables of interest. We also adjusted for the proportion of people below the poverty line, the proportion of people with 4-year degrees, and the county-level proportion of White inhabitants, the county-level 2016 Democratic Presidential vote share, and a dummy variable indicating whether a county is predominantly rural or urban. These adjustment variables were selected based on previous investigations of the distribution of hate groups[22,23].

*Adjusting for state-fixed effects.* Our primary models of the association between binding values and hate group rates address the question of whether counties that prioritize binding values tend to have higher rates of hate groups, after attempting to adjust for known risk factors and spatial autocorrelation. One potential short-coming in this analysis is that the association we observe at county-level could, in fact, be driven by unmeasured state-level confounds. That is, there could be unaccounted for state-level factors that drive the apparent association between county-level binding values and hate group rates.

This issue is often addressed via the fixed-effects estimator, which differences out group-level variance and estimates the parameter of interest as a function of within-group variance. In our case, this would involve adjusting for state-level fixed-effects and estimating the association between binding values and hate group rates from the remaining within-state variance of these variables. However, while fixed-effects estimators are widely used to adjust for unmeasured, group-level confounds, they are not a silver bullet and, in fact, they introduce their own set of issues. Fixed-effects estimators decrease statistical power[78,79], are more sensitive to measurement error than standard OLS estimators[78,79], and can wind up amplifying bias under misspecification[80].

Unfortunately, these issues are particularly relevant to our investigation of the association of county-level binding values with hate group rates. We know that hate groups are relatively rare and that the within-state variation of hate group rates is much smaller than the national county-level variance of hate group rates. We also know that our within-state county-level estimates of moral values will exhibit a bias toward their state-level mean, because these estimates were obtained via multilevel regression and poststratification. Indeed, this bias toward upper-level means is considered a feature of this estimation strategy, because it contributes regularization under sparsity. Consequently, we can expect that both the independent and dependent variables of interest will exhibit lower within-state variance. This is an issue for fixed-effects estimation, because these are exactly the conditions under which fixed-effects estimators reduce statistical power. Specifically, the degree to which fixed-effects estimators reduce power is inversely proportional to the degree to which the dependent and/or independent variables of interest vary within groups. In other words, we can reasonably infer, a priori, that the fixed-effects estimate of the association between county-level binding values and county-level hate group rates will be attenuated and have inflated standard errors[78,79]. Accordingly, null results from a fixed-effects estimator do not necessarily imply that an unmeasured confounding variable is driving the association of interest; while this is one explanation, an alternative explanation is simply that the fixed effects estimator was underpowered to detect the effect.

In addition to these issues of estimation, there is also a potential theoretical problem with using a state-level fixed effects estimator to estimate the association between county-level binding values and hate group rates: it is not necessarily clear why unmeasured state-level confounds should be expected to drive the county-level association that we observe. Fixed-effects estimators are most often applied to longitudinal panel data where observations are repeated over time within groups. For instance, a fixed-effects estimator would normally be applied to state-level panel data via the inclusion of state-level fixed effects. Thus, in a sense, states would be used as their own controls. This, however, is radically different from using a state-level fixed-effects estimator to estimate a county-level association. In that case, the validity of the procedure rests on the assumption that states are a meaningful grouping variable for counties. In many contexts, such as analyses of variables that are meaningfully driven by state-level differences, this certainly can be a valid assumption. However, this does not imply that it is always a valid assumption. Further, in the domain of binding values and hate group rates, it is not clear why states would be a natural grouping variable that must be adjusted for, especially when known risk factors are already adjusted for.

Accordingly, a state-level fixed effects estimator could enable us, in theory, to adjust for unmeasured state-level confounds and thus obtain a more accurate estimate of the parameter of interest or determine that state-level confounds are responsible for the observed county-level association. However, it could alternatively lead to an underpowered estimator with attenuated point estimates and amplified standard errors. Further, because there is little evidence that binding values or hate groups are meaningfully constrained by state boundaries, it is not even clear that state-level adjustment is theoretically valid.

Nonetheless, to directly address the possibility of state-level confounds, we report two additional negative-binomial models, one that includes state-fixed effects and one that does not. We initially attempted to estimate these models with Bayesian Spatial Filtering—as in our other analyses—in order to adjust for spatial autocorrelation; however, models estimated with state-fixed effects did not converge. Accordingly, to address the question of state- vs. county-level effects, we rely on standard general linear models estimated with maximum likelihood. Please see Table 2 in Supplementary Information for estimates of the county-level rate of hate groups with (Model 2) and without (Model 1) state-fixed effects.

## Study 2

*Participants.* To conduct this study, a sample of American participants ($N = 511$) stratified by sex (51% female), age (10–20% per each 5-year bracket ranging from 18 to 65 or older), ethnicity (62% non-Hispanic White, 17% Hispanic, 13% Black, 7% other), and political affiliation (51% Democrat) was recruited by Qualtrics Panels.

*Procedure.* After presenting a series of demographic questions, we measured participants' individualizing ($\alpha = 0.80$, 95% CI = [0.77, 0.82]) and binding ($\alpha = 0.85$, 95% CI = [0.83, 0.87]) values using the Moral Foundations Questionnaire (MFQ)[81], a 30-item scale designed to measure the degree to which people prioritize the five moral domains proposed by MFT. We then measured perceived moral wrongness using a 6-point item that asked participants to indicate, "How morally wrong is it for Muslims to spread Islamic values or laws (e.g., Sharia law) in the US instead of assimilating into American culture?"

To measure the perceived justification of EBEPs against Muslims in the US, we asked participants to imagine a man named 'Dave' who "believes Muslims are hurting his community." Participants then indicated how "justified" ('Not at all justified' = 1 to 'Extremely justified' = 7) Dave would be in committing four different EBEPs against Muslims: posting hate speech to Facebook ($M = 2.80$, Median = 2, SD = 1.88), distributing hate speech flyers ($M = 2.73$, Median = 2, SD = 1.86), yelling slurs at a Muslim ($M = 1.90$, Median = 1, SD = 1.47), and physically assaulting a Muslim ($M = 1.36$, Median = 1, SD = 1.04). These exemplar EBEPs were selected to represent a variety of potential EBEPs that are characterized by different magnitudes of social norm violation.

## Study 3

*Participants.* American participants ($N = 355$, Mean age = 33, 54% identifying as female) were recruited from Amazon Mechanical Turk and paid $1.00.

*Procedure.* Participants were asked to read a fictional news story[60] about Mexican immigrants taking jobs in Webster Springs, Illinois, a fictional town. In the news story, the Mexican immigrants were framed as undermining the local economy and, thus, harming US citizens. As in Study 2, we then asked participants to indicate on a 7-point scale (1 = 'Not at all morally wrong', 7 = 'Extremely morally wrong') the degree to which they believed it was morally wrong for Mexican immigrants to take jobs in Webster Springs ($M = 3.25$, SD = 1.81).

Prior to reading the news article, participants were also asked to complete the MFQ. Finally, participants responded to the same four EBEP items measuring the degree to which they thought EBEPs against Mexicans in Webster Springs were justified ($\alpha = 0.80$, 95% CI = [0.78, 0.83]): posting hate speech to Facebook ($M = 2.61$, Median = 2, SD = 1.70), distributing hate speech flyers in Webster Springs ($M = 2.35$, Median = 2, SD = 1.66), yelling slurs at a Mexican resident of Webster Springs ($M = 1.63$, Median = 1, SD = 1.25), and physically assaulting a Mexican resident of Webster Springs ($M = 1.17$, Median = 1, SD = 0.66).

Three participants did not complete the MFQ and an additional 28 participants spent less than 10 s reading the experimental manipulation, spent less than 8 min on the entire survey, or failed one of the MFQ manipulation checks, which were our a priori criteria to ensure data quality. Accordingly, 324 participants were retained for analysis; however, robustness checks verified that retaining these participants had no substantive effect on our results.

## Study 4

*Participants.* American participants ($N = 321$; Mean age = 33.92, SD = 10.88; 62% female) were recruited via Amazon Mechanical Turk and paid $1.00 for their participation.

*Procedure.* Participants were randomly assigned to one of two conditions—a 'high perceived moral threat' condition and a 'low perceived moral threat' condition—

**Table 1 Perceived moral wrongness of outgroup behavior by condition.**

| Condition | N | Mean | SD | Median |
|---|---|---|---|---|
| Control | 349 | 2.15 | 1.68 | 1 |
| Binding | 348 | 5.55 | 1.8 | 6 |
| Individualizing | 349 | 5.48 | 1.99 | 7 |

**Table 2 Perceived justification of EBEPs by condition.**

| Condition | Item | Mean | SD | Median |
|---|---|---|---|---|
| Control | Facebook hate speech | 2.26 | 1.82 | 1 |
| | Flyer hate speech | 2.17 | 1.77 | 1 |
| | Verbal assault | 1.97 | 1.67 | 1 |
| | Physical assault | 1.74 | 1.57 | 1 |
| Binding | Facebook hate speech | 4.5 | 2.03 | 5 |
| | Flyer hate speech | 4.24 | 2.08 | 4 |
| | Verbal assault | 3.76 | 2.09 | 4 |
| | Physical assault | 2.54 | 2.09 | 1 |
| Individualizing | Facebook hate speech | 4.86 | 1.96 | 5 |
| | Flyer hate speech | 4.79 | 1.96 | 5 |
| | Verbal assault | 3.91 | 2.1 | 4 |
| | Physical assault | 2.67 | 2.06 | 2 |

that manipulated the moral valence of a fictional outgroup. In both conditions, participants were asked to read a fictional news story similar to the story used in Study 3. However, in this version, the social outgroup was 'Sandirian'[60] immigrants. Further, in the low perceived moral threat condition, the Sandirians' actions were framed as stimulating the local economy, rather than harming it. Identical to the previous studies, we then asked participants to indicate the degree to which they believed it was morally wrong for the Sandirians to take jobs in Webster Springs ($M = 3.25$, SD = 1.81). Finally, we also asked participants to imagine a male Webster Springs resident, Dave, who believed that Sandirian immigrants were hurting his community. Participants then indicated how "justified" Dave would be in committing the EBEPs we focused on in the previous studies: posting hate speech to facebook ($M = 2.73$, Median = 2, SD = 1.87), distributing hate speech flyers in Webster Springs ($M = 2.68$, Median = 2, SD = 1.86), yelling slurs at a Sandirian resident of Webster Springs ($M = 1.80$, Median = 1, SD = 1.37), and physically assaulting a Sandirian resident of Webster Springs ($M = 1.26$, Median = 1, SD = 0.83).

Twelve participants skipped the item measuring moral wrongness and an additional 15 participants spent less than 10 s reading the experimental manipulation, which was our a priori cutoff to ensure data quality. Accordingly, 294 participants were retained for analysis, though robustness checks verified that excluding these participants had no substantive effect on our results.

**Study 5**

*Participants.* American participants ($N = 1049$, Mean age = 46.69, SD = 16.59; 50% female) were sampled via a national Qualtrics panel that was stratified with regard to age, sex, ethnicity, and political ideology and randomly assigned to the control, binding, or individualizing condition.

*Procedure.* To implement this study, we randomly assigned participants to read one of three vignettes about a fictional community, the People of the Earth, who live in Webster Springs, Florida, a fictional town. In the control condition, the People of the Earth are depicted as engaging in a morally neutral behavior, raising vegetables in a garden. In contrast, in the experimental condition designed to trigger perceptions of binding values violations, the People of the Earth are depicted as engaging in sexual rituals that involve "faeces, semen, and menstrual blood." In this experimental vignette, the outgroup's behavior was carefully designed to elicit violations of moral values oriented around purity and convention, but to also pose no direct threats to the other residents of Webster Springs. Finally, in the experimental condition targeting individualizing values, the People of the Earth are depicted as raising dogs and cats for meat. In this vignette, the outgroup's behavior was designed to elicit violations of moral values-oriented care, but, again, also to pose no direct threats to other Webster Springs residents. Importantly, in all three conditions, we clearly stated that only "some" of the community members were participating in the manipulated behavior, which means that an act of hate against a member of the community could abuse someone who has not engaged in the behavior (See vignettes in Supplementary Information Study 5; please note the study materials contained graphic content.).

After reading their assigned prompt, participants then indicated the moral wrongness of the community's behavior on a 7-point scale, from 1 = "strongly disagree" to 7 = "strongly agree" (See Table 1 for descriptive statistics). As in the previous studies, participants then reported the perceived justification of EBEPs by a Webster Springs resident against the People of the Earth (See Table 2 for descriptive statistics). Participants also reported religiosity on a 10-point scale ($M = 5.5$, SD = 3.59), ideology on a 7-point scale ($M = 3.02$, SD = 1.83), and completed the MFQ (Binding values: $M = 4.12$, SD = 0.92; individualizing values: $M = 4.56$, SD = 0.83). Twenty participants were excluded due to failing both of the MFQ attention checks, yielding $N = 1026$.

To conduct the test of moderated mediation, we estimated two Bayesian regression models. First (Model 1), we regressed z-scored PMW on a dummy variable for experimental condition (Control vs. binding values), z-scored binding values, z-scored individualizing values, and the interactions between condition and both measures of moral values. In this model, and all other models, we also adjusted for standardized self-reported religiosity, which was measured on a 0–10 scale, and 7-point self-reported political ideology ("Very liberal" to "Very conservative"). Using the same modeling strategy specified in the previous studies, we then estimated a second model (Model 2), in which the perceived justification of hate acts was regressed on all variables included in model 1, z-scored PMW, and the interaction between z-scored PMW and both measures of moral values (See Supplementary Table 21 for model estimates).

**Reporting summary**. Further information on research design is available in the Nature Research Reporting Summary linked to this article.

## Data availability

All code and publicly available data used for this research are available at https://osf.io/67cdg/. Due to privacy restrictions, we are unable to share the Pew Religious Landscape data used for our MrsP estimation procedure or the county-level moral values estimates obtained for Study 1.

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

## Acknowledgements

Work on this article was supported by grant from the National Science Foundation (BCS-1846531).

## Author contributions
J.H., M.A., and M.D. conceived the research. M.D. secured funding. J.H. and M.A. collected the data. J.H., M.A., A.M.D., B.K., G.P.-W., and L.Y. performed data analysis under M.D.'s supervision. J.H. wrote the paper. All authors provided critical edits and approved the final version of the paper.

## Competing interests
The authors declare no competing interests.
