## [Peer Review File · Nature Communications]

Editorial Note: Parts of this Peer Review File have been redacted as they concerned an analysis that was removed from the final manuscript.

Reviewers' comments:

Reviewer #1 (Remarks to the Author):

Overall, the paper provides an impressive variety of tests of the hypothesis that acts of hate tend to be morally motivated and that binding values, in particular, motivate this behavior. I think the paper makes a nice contribution, though I have some concerns about several of the studies. If the authors would discuss and clarify some of these limitations, I would be happy to see it published.

[REDACTED]

In Study 2, the authors use county-level estimates of MF endorsement to predict the prevalence of hate groups. The model is rather sparse though, controlling only for education, poverty, and size of the white population. It seems that an alternative story is that it is simply more conservative counties that have more hate groups. The authors obviously have an indicator of the political orientation of each county, since it's used in the MRP model. Why not use partisanship or ideology as a covariate? Surely other research has been done predicting county-level estimates of hate group prevalence, which could be used to inform this model.

In Study 3, the authors turn to an experiment. The authors claim to be manipulating moral wrongdoing by randomizing whether the immigrants are helping the economy or taking jobs. While this certainly has a moral component, as shown by the data, this is far from a clean manipulation. The manipulation also poses a clear and realistic threat to the economic self-interest of the local citizens. So while the data is suggestive of the authors' main hypothesis, it seems hardly conclusive. I appreciate that the authors do control for ideology, even though this is only a minor point and it is unclear how ideology was measured. It would have been great to see attention to partisan identity as well, since it is more influential on political attitudes.

In Study 4, the authors use the same design, but focus instead on Mexicans. I'm not sure that this design can tell us much about whether binding values have a unique role in hate more generally, though. The MFQ includes multiple patriotism/nationalism items, so it isn't clear whether the results are unique to binding values or are just telling us that nationalists are more likely to be anti-immigrant. Study 5 shifts the focus to Muslims, but I'm not sure this resolves the issue.

In general, the authors don't really engage with any alternative hypotheses, which would have made the results much more compelling.

Reviewer #2 (Remarks to the Author):

Non anonymous review by Jon Haidt

This is a very impressive paper. The methods are certainly impressive, although I do not understand the advanced techniques well enough to be able to pass judgment on them; I assume the editors have found at least one reviewer who is an expert in this area.

But I can comment on the central question – the role of moral motivations in acts of hate -- and its links to Moral Foundations Theory. The idea that the worst human actions--from atrocities at a mass scale down to a man beating his girlfriend for suspected infidelity—cannot be understood without taking seriously the moral motives of the perpetrator—is one of the most important ideas in the study of violence and hatred. It has been raised many times, and the authors offer a good list of citations for it, mostly from social and cultural psychology. In point #2 below I have some suggestions for improving the literature review. What the present study adds is a specific operationalization of moral motives – the “binding” foundations from MFT, which can be taken *prima facie* as expressions of moral values and motives – along with very clever methods of testing whether those moral foundations add to our explanatory ability, either of violent/hateful language, hate groups, or endorsement of hatred and violence in experiments they conducted themselves. MFT has been used often to study prejudice, but to my knowledge, MFT has not previously been used to study actual violence, hate groups, or endorsement of violence. I am particularly grateful that the authors show, in several studies, that the binding foundations add to prediction over and above other variables known to be relevant, such as political ideology (liberal to conservative). I was also just really impressed by the size and ambition of these four studies, and I found the overall psychological story being told consistently across the four studies to be very plausible.

I have three major suggestions for improvement:

1) The authors frame this paper as being important because of the rising trend of hate crimes and intergroup identity-based violence. I believe this frame should be dropped, for two reasons: 1) The overall trend over many decades is a massive drop in violence of almost all kinds, in the USA and in the world, along with a massive and steady drop in hostility and prejudice based on identity, at least within the USA. The authors should cite Steve Pinker's claims, at very least, and acknowledge that the overall trends are down, and down massively. Recent Pew data shows a continued rise in average tolerance even during the Trump years. 2) The authors discuss hate crime stats, but these are notoriously unreliable. Greg Lukianoff and I discovered this when writing chapter 6 of *The Coddling of the American Mind*. In particular, the SPLC has been exposed as a fraud, a money-making organization that does whatever it takes to perpetuate the narrative that hate is constantly on the rise. See this profile in the *New Yorker*:

<https://www.newyorker.com/news/news-desk/the-reckoning-of-morris-dees-and-the-southern-poverty-law-center>

its lists and numbers are unreliable. The authors have relied on the SPLC database for study 2; that can't be helped. But the authors should at least not refer to any claims by the SPLC in their opening framing, about how hate crimes are on the rise. Greg and I concluded that hate crimes are probably rising since 2016, but that's mostly non-violent crimes, including vandalism, and the rise is not as dramatic as is often claimed. I think that only the FBI crime victimization reports are really reliable. Everything else suffers from reporting biases.

So I'd recommend cutting the framing of "hate is rising!" and changing it to "hate is morphing – some forms are down, long term, but some are up, [REDACTED]."

2) The authors have a good paragraph listing major citations where previous scholars have written about the moral motivations of violence. I have two other books that should be added:

Baumeister (1997) *Evil*.

Stenner (2005) *the authoritarian dynamic*.

In addition, I think the paper would be much more vivid for readers if the authors could draw out one or two examples from the authors in the long list of citations. It will be so difficult for readers to grasp, intuitively, that Nazis, skinheads, and white supremacists have an intense moral worldview; readers will want to dismiss them as just overflowing with blind hatred. But the power of social psychology, like anthropology, should be to make difference intelligible. A single paragraph that began something like "For example, Fiske and Rai (2015) give the example of X, showing that the perpetrators of atrocity Y perceive that Z..." [note that it's Fiske and Rai only; Pinker just wrote a forward]. Or "As Karen Stenner (2005) showed, authoritarians are not responding to threats to themselves or their families; they are rather responding to threats to the social order...." or something more vivid than that. My point is that the paper begins with a very short and concise and abstract statement about moral motivations, then uses very advanced methods to test a complicated hypothesis. I think it is vital that the authors spend a little more time – I'd recommend two full paragraphs – helping the reader to understand the phenomenon. This is especially important so that readers can see the binding foundations as being aimed at some sort of higher good, beyond self interest, beyond blind hatred of difference.

3) This article looks only at right wing hate. As the authors acknowledge near the end, page 27, there is left-wing hate too. I think the authors should at least cite and discuss work by Jarret Crawford showing that left and right are, on some measures, equally prejudiced, its just that most of our research looks at prejudice against groups seen to be on the left, or that are favored by the left. But if you look at groups disliked by the left (such as Christians or members of the military) you find a lot of open prejudice too. I don't doubt that hatred rising to levels of violence is more common on the right, but any conservative reader will be familiar with recent cases of left wing violence (such as Antifa, and the guy who shot up the Republican softball team, including Rep. Steve Scalise). This point should be acknowledged in the intro, and perhaps returned to at the end with the suggestion that future research should examine hateful and violent rhetoric on the left.

A FEW VERY MINOR POINTS

-- [REDACTED]

-- [REDACTED]

--p 15, I'm puzzled by the graph suggesting that counties with more college grads have more hate groups. Did I read that right? Is it worth explaining why?

--on p. 19, I found myself wishing I could see a table of means. The Means are clearly very low, generally near the floor, given how nasty Dave's behavior is, and how little justification there is for it (even in the justified condition). Once again, as an intuitionist, I want to have a better feel for what's going on in the study; I don't want to just see Betas and odds ratios.

Reviewer #3 (Remarks to the Author):

This is an interesting paper that has significant strengths. The topic is important and timely. The samples and methods are impressive. Much of the paper is nicely written and many of the analyses are elegant. Despite these positive aspects, there are some problems that the authors must consider.

[REDACTED]. The second study attempts to link questionnaire responses from a website created by the founders of Morals Foundation Theory (MFT) with the presence of hate groups based on their presumed locations from the Southern Poverty Law Center. The last three studies are three separate MTurk studies that pose a hypothetical question to participants about their condoning the action of someone in a small town where an out-group is threatening to take away jobs from the community.

[REDACTED]. Study 2 is very difficult to interpret because of the quality of the two sources of data. And the third set of studies are simple but don't directly address some of the main claims of the project.

[REDACTED]

[REDACTED]

[REDACTED]

Study 2 is a nice idea in that it attempts to show a similar pattern in the geographic distribution of hate groups and separate surveys by people living in the general area of the hate groups. Each of the two data sets pose some problems. The hate group data comes from the Southern Poverty Law Center (SPLC) that for the last several years has been building a data bank of over 200 groups that they view as extremist. The overwhelming majority of the groups are right wing with clear agendas that are anti-Black, -Muslim, -Jewish, -LGBTQ+, etc. Their locations are defined where their main offices are – many of which are in larger cities (especially Washington) or in the deep south. The MFT surveys of over 100,000 people are not described so there is no sense of the demographics of the people who completed them including age, sex, percent from the deep south, etc.

One issue is that presenting data at the county level is misleading. A quick look at the observed and predicted results suggests that most of the hate groups are in southern California, Arizona, and Nevada. But these represent a very small number of counties. There are mathematically far more blue counties in the deep South – but the counties are quite small. Are the findings attributable to the fact that most binding values and most hate groups come from the deep south? Although not surprising, it would be good to know. Note that this wouldn't nullify the findings but would cast a different light on what is happening.

Studies 3-5 are quite clean and ask participants to read a very brief snippet about a mythical town in the Midwest where a large group of outsiders (Study 3 = mythical Sandarians, Study 4 = Mexicans, Study 5 = Muslims) move to town and, in the high threat condition, were described as undermining the local economy and, thus, harming "native" citizens. Participant rated if the action was morally wrong and then, in a later question, were asked to evaluate the acceptability of a hypothetical citizen, "Dave", who passed out leaflets, yelled at, or

physically harmed out-group members.

It's a nice series of studies and it would be great to see the actual means of the responses to these questions across all three studies in a single table. The one question is why are these responses threats to binding values? Doesn't this also violate a sense of fairness? How would the participants interpret the situation?

Summary

The idea of the project is quite interesting. With a major revision it could be publishable. But a revision should work to make the theory and its components more clear.

[REDACTED]

Finally, the last series of studies need to link more closely to the binding and internalizing values that are described earlier in the paper.

One final issue. Much of the recent literature on hate groups, including this paper, appears to be guided by an implicit assumption that hateful speech and acts are only perpetuated by right wing groups, ignoring that many of the ideas are true for all humans no matter what their political leanings might be. Consider today's eco-terrorists, Black nationalists, several virulent anti-Trump groups, and others. Just reading the editorials and online letters and comments of the New York Times and the Wall Street Journal demonstrates the hate language that both sides are currently spewing about each other, accusing the opposition as causing harm, cheating, betrayal, subversion, and degradation (all moral vices, I believe). It would be refreshing to see a more even-handed approach to the study of hate.

Reviewer #4 (Remarks to the Author):

This MS reports two observational and three questionnaire studies investigating the relationship between binding moral values (from Moral Foundations Theory) and hate speech/acts towards minority groups. In general, I think the MS is quite convincing and admirably combines multiple methods to make a compelling case. I do have some suggestions for improvement, which should be construed as ways to make a great paper even better.

1. The introduction argues that hate activity is on the rise. That might be true in the local sense (i.e. over the past 10 years) but in the larger sense doesn't seem right (e.g., compare to the 1960s in the US where the Klan was murdering civil rights activists). I don't think it takes away from the motivation of the paper at all to put this more accurately given the historical context (e.g., hate activity is still lower now than at many points in the past, but there seems to have been a recent uptick).

2. I understand why the authors use it, but I find the "EBEP" acronym to be sort of clunky (and I prefer to stay away from acronyms in general). If the authors can find a short but clear phrase to replace it, they should.

3. Conceptually, do the authors think of EBEPs as on a continuum with less extreme forms of prejudice, or as categorically different? If it's the former, should the psychology they describe apply to less extreme prejudice as well? One interesting wrinkle is that due to

shifting norms, what might have been “normal” prejudice 50 years ago would probably be considered hate speech now (e.g., explicit statements about the inferiority of some groups).

4. In Study 2, I am not sure (because I’m not familiar with the MrsP method) whether the analyses inherently control for county-level ideology. If they do not, I am curious to what extent county-level Binding values are associated with hate group prevalence when accounting for the partisan lean of the county.

5. In Studies 4 and 5, I would like to see descriptives (means and SDs) for the prejudice items. From Figure 5, I would guess that the mean levels of endorsement are low. I don’t think this undermines the authors’ theoretical argument, but if it is the case it should be obvious to readers and they authors might consider discussing it.

6. There seems to be a typo in the last paragraph on page 25; the sentence “the degree to which participants thought it was morally wrong for Muslims to ‘spread Islamic values’ was not positively associated with EBEP justification,” should (I think) read “was not ONLY positively associated.”

7. In the intro/discussion, the authors talk about hate crimes, but most of their data don’t speak to criminal behavior, at least not in the US. I think only the “physical assault” behavior would fall into that category, and here the authors look at ratings of justification (as opposed to, say, willingness/likelihood of engaging in it), and those ratings are low (median rating of “not at all justified”) in Study 3. The authors may want to revise their language and/or note this limitation explicitly.

Reviewer #5 (Remarks to the Author):

I was asked to take a closer look at the spatial analysis which I am happy to do. I am no subject expert on personal values and how this may help to explain aggregate level behavioral outcomes. My expertise lies more in data analysis.

I focus in my review specifically on study 2 which is based on data from an opt-in online tool.

- Biases in the measurement due to self-selection of survey participants may be corrected by relying on MrP or MrsP as shown by Wang et al. (2015) with their Xbox study. But there is no reason to believe that this is the silver bullet and we can just use any kind of data. Each dataset will require us to check anew whether the results may be representative or not. Hence, the authors need to show here that they can e.g. do a very good job in predicting democratic vote share (without using that variable as a context-level variable in the response model). If they can show that using MrP/MrsP to estimate county vote return and this is fairly accurate, then we have a reason to believe that the self-selection can be corrected for by using MrP/MrsP.

- I was wondering whether the authors used the simple synthetic or the adjusted synthetic approach by Leeman & Wasserfallen (2017). If they used the adjusted we would want to know what the basis for it was. It should not be the moral data itself but ideally stem from a sample that is closer to the random selection ideal (GSS or so).

- The authors then proceed to show that the number of hate groups per 10,000 can be explained by the local value structure. Have the authors used fixed effects and only exploit over time variation or are they rather regressing average levels on other average levels? I assume it is the latter and would like to see whether these results hold once the authors use state fixed effects and some form of controls for how rural a country is.

- I am just not sure how robust these findings are. My understanding is that the authors want to demonstrate that the aggregate outcome of hate group existence is partly explained by a specific strength of some values. I understand this to mean that average local value distribution explains the outcome of hate group existence. But to show this, we would want to see that either all people moving into these regions become more racists or that some people become radicalized and much more racists. That would constitute strong evidence for such a context-effect.

RESPONSES TO REVIEWER #1:

- 1. Overall, the paper provides an impressive variety of tests of the hypothesis that acts of hate tend to be morally motivated and that binding values, in particular, motivate this behavior. I think the paper makes a nice contribution, though I have some concerns about several of the studies. If the authors would discuss and clarify some of these limitations, I would be happy to see it published.**

Author Response: Thank you for your careful consideration of our manuscript. Your insights have helped make this a stronger contribution. We have revised our manuscript to address your comments. Below, we also discuss and clarify our position on the points that you raise.

2. [REDACTED]

Author Response: [REDACTED]

- 3. In Study 2, the authors use county-level estimates of MF endorsement to predict the prevalence of hate groups. The model is rather sparse though, controlling only for education, poverty, and size of the white population.**

Author Response: In selecting adjustment variables, we prioritized (1) matching relevant previous research and (2) minimizing the risk of overadjustment bias (Schisterman et al., 2009; York, 2018).

For instance, the models of state-level hate groups in McCann (2010) adjust for state-level non-white racial diversity, population size, education, income, and unemployment rate. Similarly, the models of county-level hate groups in Medina et al. (2018) adjust for the size of the county-level white, non-Latino population, poverty, education, population size change over time.

As you note, in our models, we control for the county-level proportions of the white population, education, and poverty. We also adjust for county-level population size with an offset variable in our regression models.

The adjustment variables we selected capture the major known factors – economic threat, racial threat, education, and population size – associated with the geographic distribution of hate group activity (e.g. those identified by relevant previous research). While there are probably many other variables that we

could have adjusted for, doing so would increase the risk of over adjustment bias (Schisterman et al., 2009) and potentially lead to less accurate estimates (York, 2018).

However, while we remain confident that these variables provide sufficiently robust adjustment for estimating our parameter of interest, we now also include two additional adjustment variables:

1. an indicator of county-level rural vs. urban status
2. The county-level 2016 Democratic Presidential vote share

Accordingly, the association we report now indicates that even after accounting for these factors, county-level variations in Binding values help explain county-level variations in hate group rates.

McCann, S. J. H. (2010). Authoritarianism, conservatism, racial diversity threat, and the state distribution of hate groups. *The Journal of Psychology*, *144*(1), 37–60.

Medina, R. M., Nicolosi, E., Brewer, S., & Linke, A. M. (2018). Geographies of Organized Hate in America: A Regional Analysis. *Annals of the Association of American Geographers. Association of American Geographers*, *108*(4), 1006–1021.

Schisterman, E. F., Cole, S. R., & Platt, R. W. (2009). Overadjustment bias and unnecessary adjustment in epidemiologic studies. *Epidemiology*, *20*(4), 488–495.

York, R. (2018). Control variables and causal inference: a question of balance. *International Journal of Social Research Methodology*, *21*(6), 675–684.

- 4. It seems that an alternative story is that it is simply more conservative counties that have more hate groups.**

Author Response: Thank you for raising this point.

This question highlights a sticky theoretical issue: given that Conservatives have been widely found to prioritize Binding values, what would the theoretical implication be if it was “simply” the case that conservative counties have more hate groups? We would argue that it is precisely Conservative counties’ higher Binding values that would drive the association between hate groups and county ideology.

Nonetheless, as noted in our answers to 4. and 6., we now also include a measure of county-level conservativeness, the county-level Democratic 2016 Presidential vote share.

- 5. The authors obviously have an indicator of the political orientation of each county, since it’s used in the MRP model. Why not use partisanship or ideology as a covariate?**

Author Response: Thank you for this question. We did not originally include partisanship/ideology in the model regression county-level hate group rates on county-level Binding values because of the conceptual confounding between Binding values and conservatism.

However, we do see the argument for including a measure of partisanship in the model and, particularly because this issue has been raised multiple times, we now include it in our model. Further, we now use a

more robust spatial regression modeling approach, Bayesian Spatial Filtering, which has been shown to be the most efficient estimator of associations between independent and dependent variables under conditions of spatial autocorrelation (Hughes, 2017; Hughes & Haran, 2013).

Hughes, J. (2017). Spatial Regression and the Bayesian Filter. In *arXiv [stat.ME]*. arXiv. <http://arxiv.org/abs/1706.04651>

Hughes, J., & Haran, M. (2013). Dimension reduction and alleviation of confounding for spatial generalized linear mixed models: Dimension Reduction and Alleviation of Confounding. *Journal of the Royal Statistical Society. Series B, Statistical Methodology*, 75(1), 139–159.

6. Surely other research has been done predicting county-level estimates of hate group prevalence, which could be used to inform this model.

Author Response: Thank you for this comment. We do indeed rely on previous research on the geographic distribution of hate groups to inform our modeling strategies (e.g. McCann, 2010; Medina et al., 2018). We also rely heavily on methodological research on estimating ecological regression models under conditions of spatial autocorrelation when the primary goal is to assess associations between independent variables and dependent variables (e.g. as opposed to the goal of prediction and forecasting; Lawson, 2013; Hughes & Haran, 2013)

Lawson, A. B. (2013). *Bayesian disease mapping: hierarchical modeling in spatial epidemiology*. CRC Press.

Hughes, J., & Haran, M. (2013). Dimension reduction and alleviation of confounding for spatial generalized linear mixed models: Dimension Reduction and Alleviation of Confounding. *Journal of the Royal Statistical Society. Series B, Statistical Methodology*, 75(1), 139–159.

McCann, S. J. H. (2010). Authoritarianism, conservatism, racial diversity threat, and the state distribution of hate groups. *The Journal of Psychology*, 144(1), 37–60.

Medina, R. M., Nicolosi, E., Brewer, S., & Linke, A. M. (2018). Geographies of Organized Hate in America: A Regional Analysis. *Annals of the Association of American Geographers. Association of American Geographers*, 108(4), 1006–1021.

7. In Study 3, the authors turn to an experiment. The authors claim to be manipulating moral wrongdoing by randomizing whether the immigrants are helping the economy or taking jobs. While this certainly has a moral component, as shown by the data, this is far from a clean manipulation. The manipulation also poses a clear and realistic threat to the economic self-interest of the local citizens. So while the data is suggestive of the authors' main hypothesis, it seems hardly conclusive.

Author's Response: Thank you for this important and constructive comment. We fully agree with your assessment of Study 3. To address this issue, we have designed a completely new study and administered it to a national Qualtrics sample (N = 1049) that is nationally representative with regard to age, gender, and ethnicity and balanced with regard to political ideology.

In this new study, we take particular care to manipulate *moral* threat in a way that does not pose a clear and realistic threat to the economic self-interest of the local citizens.

Specifically, we randomly assigned participants to either a control condition, a Binding values violation condition, or a Individualizing values violation. In each condition, participants read about a community, the People of the Earth, engaging in one of three behaviors: growing produce in a community garden (control); engaging in sex acts involving feces and bodily fluids (Binding values); or raising pet dogs and cats for meat (Individualizing values). Participants then indicated how morally wrong the community's behavior was and the degree to which acts of hate targeting a member of the community were justified.

Using this design, we now provide a more conclusive and direct test of our main hypothesis.

8. I appreciate that the authors do control for ideology, even though this is only a minor point and it is unclear how ideology was measured.

Author's Response: Thank you for noting this. We should have been more clear regarding how we measured ideology and we have revised our manuscript to fix this. Specifically, we measured ideology using a 7-point Liberal-to-Conservative identification item.

9. It would have been great to see attention to partisan identity as well, since it is more influential on political attitudes.

Author's Response: Thank you for this suggestion. We focus on ideology rather than partisan identity because political ideology is a more prominent predictor of moral judgments in the field of moral psychology. For example, most research on Moral Foundations focuses on ideology, rather than partisan identity. Further, we have no reason to expect that adjusting for partisan identity would constitute a meaningfully different adjustment with regard to our parameter of interest. Accordingly, in line with previous work, we chose to focus on ideological identification rather than partisan identity.

10. In Study 4, the authors use the same design, but focus instead on Mexicans. I'm not sure that this design can tell us much about whether binding values have a unique role in hate more generally, though. The MFQ includes multiple patriotism/nationalism items, so it isn't clear whether the results are unique to binding values or are just telling us that nationalists are more likely to be anti-immigrant. Study 5 shifts the focus to Muslims, but I'm not sure this resolves the issue.

Author's Response: Thank you for raising these issues. In our view, it seems unlikely – based on previous research – that a nationalist would not moralize issues related to the Binding values. Indeed, a nationalist is a prototypical example of someone with strong Binding values. Accordingly, similar to how it is not clear what it would mean for our effect to *merely* reflect an association between conservatism and hate, it is not clear what it would mean for the association between Binding values and hate to *merely* reflect an association with nationalism.

Nonetheless, we agree that this issue raises problems for our current studies. Our work maintains that the Moralized Threat Model is applicable across social contexts, so we should be able to show that the association between Binding values and hate is robust to contextual variation.

Importantly, we demonstrate exactly this in our new experimental study. In this study, we explicitly focus on hypothetical deviant communities that are *not* immigrant groups or groups and that are not characterized by any clear ideological associations.

11. In general, the authors don't really engage with any alternative hypotheses, which would have made the results much more compelling.

Author's Response: Thank you for this suggestion. We agree that engaging with alternative hypotheses is a powerful strategy for compelling empirical work. We now focus more comprehensively on the alternative hypothesis that political ideology might explain the association we observe between Binding values and acts of hate. In our revision, we adjust for ideology in our analysis of county-level hate groups. And, in our new study, we focus on manipulations that are not characterized by any clear ideological associations. We also now directly investigate how the implications of the Moralized Threat Hypothesis vary depending on moral domain. Specifically, we investigate whether the Moralized Threat Hypothesis generalizes equally to other moral domains (e.g. Individualizing values) or whether its implied effects are stronger in the domain of Binding values.

That said, we pair experimental and observational survey studies with analyses of real-world outcomes relevant to our hypothesis. In our view, the multi-methodological and interdisciplinary collection of studies we report in this work constitute a sufficiently compelling body of evidence. For instance, we would be less compelled by a series of survey studies that show the same effects and engage with a wider range of alternative hypotheses.

Reviewer #2 (Remarks to the Author):

Non anonymous review by Jon Haidt

- 1. This is a very impressive paper. The methods are certainly impressive, although I do not understand the advanced techniques well enough to be able to pass judgment on them; I assume the editors have found at least one reviewer who is an expert in this area.**

Author's Response: Thank you for your kind words. We sought to make this paper as methodologically robust as possible and we are grateful to see this work acknowledged.

- 2. But I can comment on the central question – the role of moral motivations in acts of hate -and its links to Moral Foundations Theory. The idea that the worst human actions--from atrocities at a mass scale down to a man beating his girlfriend for suspected infidelity— cannot be understood without taking seriously the moral motives of the perpetrator—is one of the most important ideas in the study of violence and hatred. It has been raised many times, and the authors offer a good list of citations for it, mostly from social and cultural psychology. In point #2 below I have some suggestions for improving the literature review. What the present study adds is a specific operationalization of moral motives – the “binding” foundations from MFT, which can be taken prima facie as expressions of moral**

values and motives – along with very clever methods of testing whether those moral foundations add to our explanatory ability, either of violent/hateful language, hate groups, or endorsement of hatred and violence in experiments they conducted themselves. MFT has been used often to study prejudice, but to my knowledge, MFT has not previously been used to study actual violence, hate groups, or endorsement of violence. I am particularly grateful that the authors show, in several studies, that the binding foundations add to prediction over and above other variables known to be relevant, such as political ideology (liberal to conservative). I was also just really impressed by the size and ambition of these four studies, and I found the overall psychological story being told consistently across the four studies to be very plausible.

Author's Response: Thank you for this careful review of the theoretical underpinnings of our work. We fully agree with your assessment of our research goals and we are glad that you find our literature review and research approach are sufficient and compelling.

3. **The authors frame this paper as being important because of the rising trend of hate crimes and intergroup identity-based violence. I believe this frame should be dropped, for two reasons:**
 - i. **The overall trend over many decades is a massive drop in violence of almost all kinds, in the USA and in the world, along with a massive and steady drop in hostility and prejudice based on identity, at least within the USA. The authors should cite Steve Pinker's claims, at very least, and acknowledge that the overall trends are down, and down massively. Recent Pew data shows a continued rise in average tolerance even during the Trump years.**

Author's Response: Thank you for raising this issue. We agree that it is very important to contextualize recent spikes in hate crimes and hate group activity within the larger negative trend that you describe. We now cite Pinker (2012) in our introduction and explicitly note the arguments for a global decline in prejudice and violence.

- ii. **The authors discuss hate crime stats, but these are notoriously unreliable. Greg Lukianoff and I discovered this when writing chapter 6 of *The Coddling of the American Mind*. In particular, the SPLC has been exposed as a fraud, a money-making organization that does whatever it takes to perpetuate the narrative that hate is constantly on the rise. See this profile in the New Yorker: <https://www.newyorker.com/news/news-desk/the-reckoning-of-morris-dees-and-the-southern-poverty-law-center> its lists and numbers are unreliable. The authors have relied on the SPLC database for study 2; that can't be helped. But the authors should at least not refer to any claims by the SPLC in their opening framing, about how hate crimes are on the rise. Greg and I concluded that hate crimes are probably rising since 2016, but that's mostly non-violent crimes, including vandalism, and the rise is not as dramatic as is often claimed. I think that only the FBI crime victimization reports are really reliable. Everything else suffers from**

reporting biases. So I'd recommend cutting the framing of "hate is rising!" and changing it to "hate is morphing – some forms are down, long term, but some are up, [REDACTED]."

Author's Response: Thank you for raising these excellent points and taking the time to make these suggestions for improvement.

First, we would like to clarify that we cannot find where we cite SPLC claims about hate crime rates. As evidence for recent increases in hate crime, we cite the New York Times, which report trends from the FBI Uniform Crime Reports data and Levin & Reitzel (2018), which conduct a city-level analysis in order to control for issues with reporting biases. However, we do cite the SPLC Hate Group list. While we acknowledge the issues you raise regarding this list, we feel the SPLC data offers the best opportunity to test our hypotheses to date. Accordingly, we feel that it is relevant and necessary to mention this data in our introduction.

We decided to focus on the county-level distribution of SPLC-identified hate groups because it is impossible to reliably estimate the county-level distribution of hate crimes with available data. As you note, estimating hate crime rates is very difficult. Currently, the FBI collects two types of hate crime data via the Uniform Crime Reports (UCR) and the FBI's National Crime Victimization Survey (NCVS).

The UCR is an FBI reporting program that relies on (mostly) voluntary reports from individual policing agencies. While UCR data is widely used to estimate sub-national crime rates, the possibility of differences in reporting standards across agencies raise very serious issues for estimates of sub-national hate crime rates. This issue is particularly relevant for our work: we would expect policing agencies to place *less* focus on hate crimes (e.g. less training, less reporting) in areas where hate crimes might be seen as more morally acceptable. This kind of regional reporting bias could completely mask a very real association between Binding values and hate crimes. In addition to the issue of agency reporting biases, there is also the issue of victim reporting biases. Research suggests that the majority of bias-motivated crimes are not reported to police (Wilson, 2014; Pezzella, Fetzer, Keller, 2019). Further it seems reasonable to expect that the rate of non-reporting might vary geographically according to factors such as police receptivity to hate crime reporting and local attitudes toward the victim's targeted identity group. Again, this kind of reporting bias could completely mask a real association between Binding values and hate crime.

In contrast to the UCR, the NCVS data is collected via a random, nationally representative sample. Accordingly, it does not suffer from the same kinds of severe reporting biases that characterize the UCR. However, access to geolocated NCVS data is highly restricted and requires an extremely rigorous, multi-year review process. Indeed, while there is some work focused on small-area crime estimation using the NCVS, we are not aware of *any* work in the domain of hate crime. Accordingly, while an unbiased estimate of county-level hate crime rates could conceivably be estimated from the NCVS, taking this approach was not feasible.

For these reasons, we decided to focus on the county-level distribution of hate groups. As you note, there are many problems with the SPLCs list of hate groups; however, because the SPLC does not rely on agency level reports, there is no reason to expect the kind of geographic reporting biases that characterize the UCR data.

That said, if the SPLC has an extreme liberal bias, one might expect that the locations of SPLC-identified hate groups might be correlated with regional political ideology. That is, if the SPLC is primarily oriented around the task of demonizing conservative groups in order to raise funding, one might reasonably expect the organizations they label as hate groups to be spatially correlated with conservativeness.

We agree that this is certainly an issue for our analysis and we now include county-level 2016 Democratic Presidential vote share and county-level rural vs. urban status.

Importantly, after accounting for rural status and county-level partisanship, we still observe a robust association between county-level Binding values and the county-level rate of hate groups.

Of course, this does not address the issues of whether the SPLC is fraudulent or their list of hate groups unreliable. Regarding that issue, we fully agree that there are likely serious issues with the SPLC's reporting methods. However, to our knowledge, there is not consensus on the view that the SPLC is a fraud that "does whatever it takes to perpetuate the narrative that hate is constantly on the rise." For example, see this Washington Post article (<https://www.washingtonpost.com/news/magazine/wp/2018/11/08/feature/is-the-southern-poverty-law-center-judging-hate-fairly/>).

But, perhaps more importantly, if the SPLC list of hate groups was extremely unreliable, what would this mean for our analysis? One reasonable expectation might be that we would observe no association between Binding values and the rate of hate groups, especially after controlling for what seem to be likely confounds (e.g. variables that a liberal-biased organization might condition on when identifying groups to target), such as rural status and county-level vote share. However, even after controlling for these factors, we still observe the hypothesized association.

Finally, while we acknowledge the many measurement issues with UCR hate crime data, it is also worth noting that recent research has found positive associations between regional variations in UCR reported hate crime and the geographic distribution SPLC-identified hate groups (Jendryke & McClure, 2019). Similarly, research has also found positive geographic associations between far-right ideologically motivated homicides recorded in the U.S. Extremist Crime Database and the distribution of SPLC-identified hate groups (Adamczyk, Gruenwald, Chermak, & Freilich, 2014). Notably, these are exactly the kinds of associations one would expect if the SPLC data was at least moderately reliable.

Of course, there are many other potential issues that could be raised with regard to the SPLC data and our reliance on it to test our hypothesis. This is exactly why we paired this study with another observational study that is free from these kinds of reporting biases (Study 1) and a series of experimental and observational survey studies. While the association we observe in Study 2 could be spuriously caused by unreliable reporting from the SPLC, we think that this is unlikely, given that we find comparable effects across a range of other methodologies and data sources.

Finally, regarding the question of whether hate crime has increased in recent years, there seems to be little evidence to the contrary. As evidence for this claim, we cited multiple sources, none of which were the SPLC. However, to further bolster this claim, we now also cite Levin and Reitzel (2018), which found a 12.5% increase in hate crime in 2017, despite a small decrease in crime not motivated by bias. Finally, we would hesitate to dismiss non-violent hate crimes when considering the overall trend of hate crime. While they are certainly less extreme than violent hate crimes, they can have substantive negative effects on their victims and victims' communities.

That said, again, we want to acknowledge that we share your concern about the validity of the SPLC hate group list. Given recent news coverage and criticism of the SPLC, we agree that it is likely that the SPLC

list is biased and perhaps contaminated with a higher false positive rate than would be ideal for our analysis. However, in the context of our entire program of research, we believe that this study still provides useful convergent evidence for our hypotheses.

Also, you make a very good point that the recent increase in hate crimes must be contextualized within the larger negative trend of global violence. Accordingly, we now note this larger negative trend in our introduction.

Adamczyk, A., Gruenewald, J., Chermak, S. M., & Freilich, J. D. (2014). The Relationship Between Hate Groups and Far-Right Ideological Violence. *Journal of Contemporary Criminal Justice*, 30(3), 310–332.

Jendryke, M., & McClure, S. C. (2019). Mapping crime--Hate crimes and hate groups in the USA: A spatial analysis with gridded data. *Applied Geography*, 111, 102072.

Levin, B., & Reitzel, J. D. (2018). Hate crimes rise in US cities and counties in time of division and foreign interference. *Rep. Nation, Cent. Study Hate Extremism, Calif. State*.

Pezzella, F. S., Fetzer, M. D., & Keller, T. (2019). The Dark Figure of Hate Crime Underreporting. *The American Behavioral Scientist*, 0002764218823844.

Wilson, M. M. (2014). Hate crime victimization, 2004-2012: Statistical tables. Retrieved from <http://www.bjs.gov/index.cfm?ty=pbdetail&iid=4883>

4. The authors have a good paragraph listing major citations where previous scholars have written about the moral motivations of violence. I have two other books that should be added: Baumeister (1997) Evil. Stenner (2005) the authoritarian dynamic.

Author's Response: Thank you for suggesting these citations, we have added them to our manuscript.

5. In addition, I think the paper would be much more vivid for readers if the authors could draw out one or two examples from the authors in the long list of citations. It will be so difficult for readers to grasp, intuitively, that Nazis, skinheads, and white supremacists have an intense moral worldview; readers will want to dismiss them as just overflowing with blind hatred. But the power of social psychology, like anthropology, should be to make difference intelligible. A single paragraph that began something like “For example, Fiske and Rai (2015) give the example of X, showing that the perpetrators of atrocity Y perceive that Z...” [note that it’s Fiske and Rai only; Pinker just wrote a forward]. Or “As Karen Stenner (2005) showed, authoritarians are not responding to threats to themselves or their families; they are rather responding to threats to the social order.....” or something more vivid than that. My point is that the paper begins with a very short and concise and abstract statement about moral motivations, then uses very advanced methods to test a complicated hypothesis. I think it is vital that the authors spend a little more time – I’d recommend two full paragraphs – helping the reader to understand the phenomenon. This is especially important so that readers can see the binding foundations as being aimed at some sort of higher good, beyond self interest, beyond blind hatred of difference.

Author's Response: Thank you for this excellent suggestion! We have followed your advice and have added two paragraphs discussing the link between morally motivated violence and morally motivated acts of hate.

6. This article looks only at right wing hate. As the authors acknowledge near the end, page 27, there is left-wing hate too. I think the authors should at least cite and discuss work by Jarret Crawford showing that left and right are, on some measures, equally prejudiced, its just that most of our research looks at prejudice against groups seen to be on the left, or that are favored by the left. But if you look at groups disliked by the left (such as Christians or members of the military) you find a lot of open prejudice too. I don't doubt that hatred rising to levels of violence is more common on the right, but any conservative reader will be familiar with recent cases of left wing violence (such as Antifa, and the guy who shot up the Republican softball team, including Rep. Steve Scalise). This point should be acknowledged in the intro, and perhaps returned to at the end with the suggestion that future research should examine hateful and violent rhetoric on the left.

Author's Response: Thank you for raising this issue. We fully agree that left-wing hate exists, too, and, indeed, we hope to focus on left wing hate in future research. We now more explicitly note the presence of left wing hate in our manuscript.

That said, we do want to emphasize that our current work focuses on *hatred* and not *prejudice*. While we would not be surprised to find that liberals are just as prejudiced toward some groups as many conservatives, we are unaware of any evidence suggesting that liberals are as likely to express their prejudice via extreme behaviors such as hate crime. At least in recent years, far-rights (rather than far-lefts) have held the market share in hate-motivated terrorist attacks. And, year after year, a disproportionate number of hate crimes are perpetrated against classes of people that are conventionally disliked by conservatives (and not liberals). Finally, regardless of the SPLC's taxonomy, we suspect that there are far more violent right-wing hate groups than violent left-wing hate groups.

Given all of this, we believe that the issue of right-wing acts of hate is currently more pressing. However, we fully believe that in a different socio-cultural moment, left-wing acts of hate could very well supersede right-wing acts of hate. Indeed, while we focus on right-wing acts of hate, the Moralized Threat Model is generally agnostic to political ideology – we propose that perceived moral violations trigger acts of hate and we expect that this effect applies just as strongly to liberals as it does to conservatives.

7. A FEW VERY MINOR POINTS

[REDACTED]

Author's Response: [REDACTED]

- d. --p 15, I'm puzzled by the graph suggesting that counties with more college grads have more hate groups. Did I read that right? Is it worth explaining why?

Author's Response: We were also puzzled by this positive association. While we suspect that the answer to this question would be interesting, we ultimately decided that investigating it was beyond the scope of this work and felt that it would be best to not speculate on possible explanations. That said, it could be that hate groups are more prevalent in areas with higher levels of political polarization or economic inequality; if this were the case, one might expect a polynomial association between the rate of college graduates and rate of hate groups. Of course, there are many other possible explanations for the association.

- e. --on p. 19, I found myself wishing I could see a table of means. The Means are clearly very low, generally near the floor, given how nasty Dave's behavior is, and how little justification there is for it (even in the justified condition). Once again, as an intuitionist, I want to have a better feel for what's going on in the study; I don't want to just see Betas and odds ratios.

Author's Response: Thank you for this suggestion. We fully agree that understanding the implications of our study requires an understanding of the response distributions for the EBEP items. For each study, we now specify the means, standard deviations, and medians for all four EBEP items.

Reviewer #3 (Remarks to the Author):

1. This is an interesting paper that has significant strengths. The topic is important and timely. The samples and methods are impressive. Much of the paper is nicely written and many of the analyses are elegant. Despite these positive aspects, there are some problems that the authors must consider.

Author Response: Thank you for this feedback. Your comments were very helpful and we believe our paper is stronger now that we have addressed them.

2. [REDACTED]. The second study attempts to link questionnaire responses from a website created by the founders of Morals Foundation Theory (MFT) with the presence of hate groups based on their presumed locations from the Southern Poverty Law Center. The last three studies are three separate MTurk studies that pose a hypothetical question to participants about their condoning the action of someone in a small town where an out-group is threatening to take away jobs from the community.

Author Response: Thank you for taking the time to read each study! We would like to note, however, that one of the survey studies actually used participants recruited by Qualtrics Inc and, as such, was not an

MTurk study. We only note this in order to emphasize that we sought to provide convergent evidence across sample populations.

- 3. [REDACTED]. Study 2 is very difficult to interpret because of the quality of the two sources of data. And the third set of studies are simple but don't directly address some of the main claims of the project.**

Author Response: Thank you for these comments. Since you address these items in detail below, we have reserved the substance of our responses for your more detailed feedback. That said, while we agree that Study 2 does have shortcomings, we strongly disagree that it is very difficult to interpret. We made an extensive effort to account for a range of issues and that work is comparable to other peer-reviewed research involving small-area estimation and geographic studies of the distribution of hate groups. Further, we have added an additional experimental study to address the issues you raise regarding studies 3-5.

- 4. [REDACTED].**

Author Response: [REDACTED]

- 5. [REDACTED].**

Author Response: [REDACTED]

- 6. [REDACTED].**

Author Response: [REDACTED]

- 7. Study 2 is a nice idea in that it attempts to show a similar pattern in the geographic distribution of hate groups and separate surveys by people living in the general area of the hate groups. Each of the two data sets pose some problems. The hate group data comes from the Southern Poverty Law Center (SPLC) that for the last several years has been building a data bank of over 200 groups that they view as extremist. The overwhelming majority of the groups are right wing with clear agendas that are anti-Black, -Muslim, -Jewish, -LGBTQ+, etc. Their locations are defined where their main offices are – many of which are in larger cities (especially Washington) or in the deep south. The MFT surveys of over 100,000 people are not described so there is no sense of the demographics of the people who completed them including age, sex, percent from the deep south, etc.**

Author Response: Thank you for taking the time to note these details. We now include demographic information about the MFT respondents.

- 8. One issue is that presenting data at the county level is misleading. A quick look at the observed and predicted results suggests that most of the hate groups are in southern California, Arizona, and Nevada. But these represent a very small number of counties. There are mathematically far more blue counties in the deep South – but the counties are quite small. Are the findings attributable to the fact that most binding values and most hate groups come from the deep south? Although not surprising, it would be good to know. Note that this wouldn't nullify the findings but would cast a different light on what is happening.**

Author Response: Thank you for raising these questions. Unfortunately, we do not quite follow this argument. Further, it is not clear to us how our county-level analysis could be “misleading.” A state-level analysis would have obscured substantial sub-state variation and, we would argue, ultimately would have provided a less informative perspective on the association between Binding values and hate groups.

Further, there are clusters of hate groups in both the Deep South and in the Southwest, as well as in New England. Given that we originally adjusted for the most likely confounds (education, poverty, and ethnic composition) and now include additional adjustments for county-level ideology and rural vs. urban, it is not quite clear to us how we should interpret geographic covariation between Binding values and hate groups if not as evidence for an association.

Finally, during our revision process, we did estimate additional models that adjusted for the expected state-level rate of hate groups. In these models, the coefficient for Binding values reflected the effect on hate group rate of Binding values *after adjusting* the average rate of hate groups at the state level.

- 9. Studies 3-5 are quite clean and ask participants to read a very brief snippet about a mythical town in the Midwest where a large group of outsiders (Study 3 = mythical Sandarians, Study 4 = Mexicans, Study 5 = Muslims) move to town and, in the high threat condition, were described as undermining the local economy and, thus, harming “native” citizens. Participant rated if the action was morally wrong and then, in a later question, were asked to evaluate the acceptability of a hypothetical citizen, “Dave”, who passed out leaflets, yelled at, or physically harmed out-group members.**

Author Response: Thank you for taking the time to note these details.

- 10. It’s a nice series of studies and it would be great to see the actual means of the responses to these questions across all three studies in a single table.**

Author Response: Thank you for this request. We now describe the distributions (means, SDs, and medians) for each EBEP item in each study in the main manuscript. As the distributions are quite consistent across studies and our SM is already very long, we declined to add an additional table to SM.

- 11. The one question is why are these responses threats to binding values? Doesn’t this also violate a sense of fairness? How would the participants interpret the situation?**

Author Response: Thank you for this question. In general, the Binding values are oriented around threats to group cohesion, security, and traditions. In studies 3-5, we focus on two general out-group behaviors: undocumented immigrants – specifically, a fictional immigrant group (Study 3) and Mexicans (Study 4) – taking jobs and Muslim immigrants spreading “Islamic values” (Study 5).

First, we would like to note that we fully agree that undocumented immigrants taking jobs can be understood as an Individualizing (e.g. fairness) violation. Indeed, this is why we also focused on Muslims spreading “Islamic values,” as this is difficult to understand as anything other than a threat to ingroup norms and traditions.

Nonetheless, in our view, it would be reductive to construe undocumented immigrants taking jobs as merely a fairness violation because it also involves violations of group norms around hierarchy and order. Consistent with this, in study 4 we find that even after controlling for individual-level political ideology, people with stronger Binding values are more likely to believe that it is morally wrong for undocumented Mexicans to take jobs, compared to people with weaker Binding values. Further, no such positive association is found for Individualizing values. This suggests that the behavior is more morally triggering for people who prioritize Binding values (and not those who prioritize Fairness values).

Finally, to more thoroughly address this issue, we conducted a new experimental study (Study 6 in our revised manuscript). In this study, we directly target Binding values through an experimental manipulation that depicts the social outgroup as engaging in taboo sexual rituals. Importantly, we show

that participant's Binding values moderates the effect of experimental condition, such that people low on Binding values were, on average, far less likely to think the outgroup's behavior was immoral and that acts of hate against the group were justified. In other words, in our new study, we show that Binding values function like a treatment susceptibility factor that moderates people's responses to Binding values violations.

12. Summary

- a. **The idea of the project is quite interesting. With a major revision it could be publishable. But a revision should work to make the theory and its components more clear. [REDACTED]. Finally, the last series of studies need to link more closely to the binding and internalizing values that are described earlier in the paper.**

Author Response: Thank you for highlighting these issues. [REDACTED]. Regarding the survey studies, we have clarified our discussion of Moral Foundations Theory throughout the manuscript in order to more clearly link these studies to the Individualizing and Binding values. We have also added an additional study that offers a more direct test of the Moralized Threat Hypothesis. In this study, we manipulate both Binding and Individualizing values and find evidence for the following hypotheses: 1. experimentally manipulated Binding values violations cause increased beliefs in the justification of hate acts; 2. the degree to which the violations are seen as morally wrong mediates this association; 3. Binding values moderates the mediation effect.

- b. **One final issue. Must of the recent literature on hate groups, including this paper, appears to be guided by an implicit assumption that hateful speech and acts are only perpetuated by right wing groups, ignoring that many of the ideas are true for all humans no matter what their political leanings might be. Consider today's eco-terrorists, Black nationalists, several virulent anti-Trump groups, and others. Just reading the editorials and online letters and comments of the New York Times and the Wall Street Journal demonstrates the hate language that both sides are currently spewing about each other, accusing the opposition as causing harm, cheating, betrayal, subversion, and degradation (all moral vices, I believe). It would be refreshing to see a more even-handed approach to the study of hate.**

Author Response: Thank you for raising this point. Several readers have raised the point that our manuscript focuses on acts of hate perpetrated by right-wing groups and, consequently, neglects acts of hate perpetrated by left-wing groups. First, we would like to emphasize that the Moralized Threat Hypothesis is consistent with the belief that any group of any political ideology can engage in acts of hate under the right conditions. And we fully recognize that left-wing hatred also occupies an increasingly prominent space in American society.

Nonetheless, it is also the case that in our current socio-cultural moment, more hate crimes and terrorist attacks are perpetrated by people who subscribe to right-wing ideologies than left-wing ideologies and far

more hate groups espouse right-wing ideologies than left-wing ideologies. That is, there is no left-wing equivalents of the Klu Klux Klan, Aryan Brotherhood, or American Nazi Party. In our view, this does not mean that left-wing hate is not a current issue, but it does suggest that right-wing hate may be more a widespread issue. And one consequence of this is that it is easier to study right-wing hate because it is better documented and easier to access.

That said, the reviewer's point is well taken. As we note above, we now offer a more detailed discussion of how hate can arise on both sides of the aisle.

Reviewer #4 (Remarks to the Author):

This MS reports two observational and three questionnaire studies investigating the relationship between binding moral values (from Moral Foundations Theory) and hate speech/acts towards minority groups. In general, I think the MS is quite convincing and admirably combines multiple methods to make a compelling case. I do have some suggestions for improvement, which should be construed as ways to make a great paper even better.

Author Response: Thank you for your positive words and thoughtful suggestions.

- 1. The introduction argues that hate activity is on the rise. That might be true in the local sense (i.e. over the past 10 years) but in the larger sense doesn't seem right (e.g., compare to the 1960s in the US where the Klan was murdering civil rights activists). I don't think it takes away from the motivation of the paper at all to put this more accurately given the historical context (e.g., hate activity is still lower now than at many points in the past, but there seems to have been a recent uptick).**

Author Response: Thank you for this suggestion. We have modified the framing we use in the introduction to more accurately contextualize the recent upward trends in hate acts.

- 2. I understand why the authors use it, but I find the "EBEP" acronym to be sort of clunky (and I prefer to stay away from acronyms in general). If the authors can find a short but clear phrase to replace it, they should.**

Author Response: Thank you for this suggestion. While we generally share your feelings toward acronyms, we put a fair amount of thought into the label for our construct of interest. Extreme behavioral expressions of prejudice are exactly what we aim to study and explain in this work. We also considered phrases like "acts of hate," but we feel reference to "hate" opens up questions about how to operationalize and classify "hate" and "hatred." As those questions are not central to our focus here, we felt that reference to hatred could distract from our primary aims. Accordingly, in this case we feel that it is better to suffer a little clunkiness in order to prioritize linguistic precision.

- 3. Conceptually, do the authors think of EBEPs as on a continuum with less extreme forms of prejudice, or as categorically different? If it's the former, should the psychology they describe apply to less extreme prejudice as well? One interesting wrinkle is that due to**

shifting norms, what might have been “normal” prejudice 50 years ago would probably be considered hate speech now (e.g., explicit statements about the inferiority of some groups).

Author Response: This is a very interesting question. Unfortunately, our current program of research does not directly address this question, so we would hesitate to take a strong position on any possible answer. On one hand, it is easy to imagine how hatred might emerge from what is initially just mild dislike for a social group. Further, it seems likely that less extreme forms of prejudice might serve as a foundation for hatred. In this sense, it might be fair to place acts of hate on a continuum of prejudice where “mild” prejudice or dislike constitutes the opposite pole. However, we also suspect that the psychological/neurological profile of prejudice likely changes as one moves across this spectrum. For instance, hatred involves extreme affective responses that are most likely not involved in milder forms of prejudice. So, in this sense, we suspect that there may also be qualitative differences in the psychological/neurological components involved in weaker forms of prejudices vs. extreme behavioral expressions of prejudice.

- 4. In Study 2, I am not sure (because I’m not familiar with the MrsP method) whether the analyses inherently control for county-level ideology. If they do not, I am curious to what extent county-level Binding values are associated with hate group prevalence when accounting for the partisan lean of the county.**

Author Response: Thank you for this comment. We now adjust for county-level Democratic vote share in our primary analysis. Notably, adjusting for this factor did not substantively change our findings.

- 5. In Studies 4 and 5, I would like to see descriptives (means and SDs) for the prejudice items. From Figure 5, I would guess that the mean levels of endorsement are low. I don’t think this undermines the authors’ theoretical argument, but if it is the case it should be obvious to readers and they authors might consider discussing it.**

Author Response: Thank you for this suggestion. We now include means, SDs, and medians for the EBEP items in all studies.

- 6. There seems to be a typo in the last paragraph on page 25; the sentence “the degree to which participants thought it was morally wrong for Muslims to ‘spread Islamic values’ was not positively associated with EBEP justification,” should (I think) read “was not ONLY positively associated.”**

Author Response: Thank you for highlighting this error. We have fixed it in our revised manuscript.

- 7. In the intro/discussion, the authors talk about hate crimes, but most of their data don’t speak to criminal behavior, at least not in the US. I think only the “physical assault” behavior would fall into that category, and here the authors look at ratings of justification (as opposed to, say, willingness/likelihood of engaging in it), and those ratings are low (median rating of “not at all justified”) in Study 3. The authors may want to revise their language and/or note this limitation explicitly.**

Author Response: Thank you for this suggestion. While we agree that our data do not directly speak to behavior that would be considered criminal in the U.S., we think that it is nonetheless important to explicitly highlight the kinds of behaviors that we classify as extreme behavioral expressions of prejudice. Accordingly, to address your concern, when we describe our current work in the Introduction, we now explicitly note the EBEPs we focus on: hate speech and hate group activity.

Reviewer #5 (Remarks to the Author):

- 1. I was asked to take a closer look at the spatial analysis which I am happy to do. I am no subject expert on personal values and how this may help to explain aggregate level behavioral outcomes. My expertise lies more in data analysis.**

Author Response: Thank you for this clarification and for your thorough review of our paper.

- 2. Biases in the measurement due to self-selection of survey participants may be corrected by relying on MrP or MrsP as shown by Wang et al. (2015) with their Xbox study. But there is no reason to believe that this is the silver bullet and we can just use any kind of data. Each dataset will require us to check anew whether the results may be representative or not. Hence, the authors need to show here that they can e.g. do a very good job in predicting democratic vote share (without using that variable as a context-level variable in the response model). If they can show that using MrP/MrsP to estimate county vote return and this is fairly accurate, then we have a reason to believe that the self-selection can be corrected for by using MrP/MrsP.**

Author Response: Thank you for this comment. We completely agree that MrP/MrsP is not a silver bullet and that validation is an essential component in any downstream analysis that relies on MrP/MrsP estimates. We should have included a validity check in our initial manuscript and the fact that we didn't was an oversight on our part. So, thank you for bringing this up.

Our revised manuscript now includes results from a validation study in Study 2 Supplemental Material. In this study, we estimate the county-level population proportion that identifies as conservative (i.e. slightly conservative, conservative, or very conservative) by applying MrsP to the Moral Foundations dataset we obtained from YourMorals.org.

While it would have been more conventional, with regard to the MrP literature, to estimate the county-level Republican or Democratic vote share, this was not an option for us because we do not have sufficient data for respondents' voter intentions or behavior. As such, one drawback to our validity test is that there is an unknown ceiling for the association between the county-level proportion of people who identify as at least slightly conservative and the county-level Republican vote share. That is, we know that there is not a 1:1 relationship between ideological identification and voter behavior, so the association between a "perfect" MrsP estimate of county-level conservativeness would not necessarily be perfectly correlated with county-level Republican vote share.

Nonetheless, left-right orientation is one of the strongest predictors of vote choice (Jou & Dalton, 2017) and the county-level proportion of people who identify as at least slightly conservative should thus be strongly associated with the county-level Republican vote share.

Accordingly, if the MrsP procedure we rely on to estimate county-level moral values sufficiently accounts for response-biases, then we should be able to detect a strong association between estimates of the county-level proportion of self-identified conservatives and the observed county-level Republican vote share.

To conduct our validity test, we estimated the county-level proportion of self-identified conservatives using the MrsP procedure that we use to estimate county-level moral values. However, one key difference is that we *did not* include any context-level measures of political ideology or vote share in the MrsP response model. More specifically, at the individual-level, the response model adjusted for respondent ethnicity, gender, level of education, age, and frequency of religious attendance (these variables were identical to those we include in the response models used to estimate county-level moral values). At the context-level, we included the county-level rate of protestant evangelicals, which we obtained from the 2010 Religious Census (Grammich, 2012).

Finally, to investigate the degree to which MrsP can correct for self-selection bias in the YourMorals data, we evaluated the association between county-level 2012 and 2016 Presidential election Republican vote shares and our estimates of county-level conservatism.

We found that both 2012 and 2016 county-level Republican vote shares were strongly correlated with our estimate of county-level proportions of conservatives, $r(3,104) = .60, p < .01, 95\% CI = [0.58, 0.62]$ and $r(3,104) = .63, p < .01, 95\% CI = [0.60, 0.65]$, respectively. Notably, we also found that while the 2012 and 2016 Root Mean Square Errors were relatively high (2012 RMSE = 0.17 and 2016 RMSE = 0.20), systematic bias, e.g. Mean bias Error (MBE), accounted for a substantial portion of this error (2012 MBE = -0.12 and 2016 MBE = -0.16). In other words, for both elections, the estimated county-level proportion of conservatives was, on average, systematically lower than the Republican vote share. To estimate RMSE with systematic bias removed, we calculated

$$SD = \sqrt{RMSE^2 - MBE^2}.$$

SD for both 2012 and 2016 county-level Republican vote share was 0.12.

Finally, we also examined the association between estimated county-level proportions of conservatives and Root Square Error (RSE) and Bias Error (BE) in order to evaluate the degree to which these sources of error are constant across the scale of our estimates. We found no evidence for an meaningful association between the estimated county-level proportion of conservatives and RSE and BE with regard to the 2012 Republican vote share, $r(3,104) = -0.001, p = .93, 95\% CI = [-0.04, 0.03]$ and $r(3,104) = .03, p = 0.11, 95\% CI = [-0.01, 0.06]$. In contrast, we did observe a weak association with RSE and BE with regard to the 2016 Republican vote share, $r(3,104) = 0.05, p < 0.01, 95\% CI = [0.02, 0.09]$ and $r(3,104) = -0.04, p = 0.01, 95\% CI = [-0.07, -0.01]$. That is, RSE is expected to be slightly higher for counties with larger estimated proportions of conservatives and, further, the degree to which the estimated proportion of conservatives systematically underestimates the 2016 Republican vote share (e.g. the magnitude of the negative bias of our estimate) increases slightly as the estimated proportion of conservatives increases. Consistent with these findings, the correlation between the observed errors and errors expected under assumptions of normality were 0.99 and 0.98 for election years 2012 and 2016.

In conclusion, we found that our estimates of county-level proportions of conservatives were strongly associated with county-level Presidential Republican vote shares from the 2012 and 2016 elections. We also found that our estimates of county-level proportions of conservatives were systematically lower than observed county-level Republican vote shares. However, this is not particularly surprising, as some portion of self-identified moderates in each county likely voted for the Republican candidate. Accordingly, it is reasonable to expect a priori that our estimate of the county-level proportion of conservatives would be lower than the county-level Republican vote share. Finally, we also found evidence that suggests that our estimates were comparably accurate across their scale.

All together, these results suggest that our MrsP model sufficiently accounts for response biases in the YourMorals data and that estimates derived from this model can be reasonably expected to approximate population values.

Jou, W., & Dalton, R. J. (Eds.). (2017). Left-Right Orientations and Voting Behavior. In *Oxford Research Encyclopedia of Politics*. Oxford University Press.

- 3. I was wondering whether the authors used the simple synthetic or the adjusted synthetic approach by Leeman & Wasserfallen (2017). If they used the adjusted we would want to know what the basis for it was. It should not be the moral data itself but ideally stem from a sample that is closer to the random selection ideal (GSS or so).**

Author Response: Thank you for this question! We used Leeman & Wasserfallen's (2017) adjusted synthetic approach and we derived the adjustments from Pew's 2014 Religious Landscape Survey (RLS; Hackett et al., 2014). We selected the RLS instead of the GSS, ANES, or other common national samples for two primary reasons. First, the RLS ($N \sim 35,000$) is much larger than the GSS or ANES (single waves $\sim 2,000-4000$) and thus should yield more precise estimates of the demographic joint distribution. Second, the RLS sample is nationally representative and was recruited via random digit dialing, which is preferable to the cluster-sampling procedure used for the GSS and ANES.

This is now clarified in our manuscript.

Hackett, C., Grim, B., Stonawski, M., Skirbekk, V., Kuriakose, N., & Potančoková, M. (2014). Methodology of the pew research global religious landscape study. In *Yearbook of International Religious Demography 2014* (pp. 167-175). Brill.

- 4. The authors then proceed to show that the number of hate groups per 10,000 can be explained by the local value structure. Have the authors used fixed effects and only exploit over time variation or are they rather regressing average levels on other average levels?**

Author Response: Thank you for these questions. If we understand you correctly, your assumption is correct. Our dependent variable is the total number of hate groups reported by the SPLC from 2012-2015 aggregated to the county-level. Our independent variables, which are all time invariant, are MrsP estimated county-level Binding and Individualizing values and a set of county-level adjustment variables. In other words, our observations are at the county-level, such that we have a single set of time invariant measurements for each county.

Perhaps this is confusing because our dataset spans several years and this raises the question of why not disaggregate into county x years and conduct our analysis at that level. The problem with this approach is that it would require yearly estimates of county-level moral values, which we do not have. While our YourMorals sample does indeed span 5 years, we use the full sample to estimate time invariant county-level moral values. If, instead, we tried to estimate county-level moral values for each year, those estimates would be based on approximately 20,000 observations each, assuming a uniform response distribution over years. Given that there are 3000+ counties in the contiguous U.S., this would induce an enormous amount of sparsity and would likely yield estimates that are unreliable and/or too severely smoothed by the MrsP model (i.e. a large portion of counties would have few to none observations).

Accordingly, while it would certainly be valuable to exploit temporal variation in our analysis, such an approach is not feasible because our moral values estimates are time invariant.

5. I assume it is the latter and would like to see whether these results hold once the authors use state fixed effects and some form of controls for how rural a country is.

Author Response: Thank you for taking the time to think through our analysis and offer these suggestions for improvement. We fully agree that adjusting for rural vs. urban is a good idea and we now include that in our model.

However, we do not follow the rationale for adding state fixed effects to our model. Instead of improving our estimates of the association between county-level Binding values and hate group rates, adding state fixed effects in our model would fundamentally change the question addressed by the model.

Our goal is to estimate the association between county-level Binding values and the county-level rate of hate groups. Under our current model specification, the regression coefficient estimated for Binding values represents this association, assuming all confounding variables are also included in the model.

If we were to add state fixed effects to our model, we would no longer be estimating the association between county-level Binding values and hate group rates. Instead, we would be estimating this association *after adjusting for state-level differences*. In other words, a model with state fixed effects would ask the question: do county-level Binding values explain variation in the county-level rate of hate groups *within a given state*.

This model poses several problems. First, it does not address our question of interest. Second, its validity rests on the assumption that state boundaries encode meaningful/relevant hierarchical structure with regard to the county-level rate of hate groups. In other words, the only reason to adjust for state fixed effects is to account for unmeasured confounding factors that operate at the state level. In many research settings, this a reasonable and often necessary approach. However, in our case, it is not at all clear why *state-level differences* might confound the association between county-level moral values and the county-level rate of hate groups. Without a reasonable argument for mechanism, including state fixed effects would raise the risks of over adjustment and adjustment bias (Schisterman, Cole, & Platt, 2009; Breslow, 1982).

Of course, one could argue that if there are no confounding state-level effects, then estimates of the association between Binding values and hate group rates should be robust to including state fixed effects. However, this is not necessarily true, because, again, adding state fixed effects to our model would fundamentally change the question addressed by the model. With state fixed effects, our analysis would focus on whether within-state variation in county-level Binding values explains within-state variation in

the county-level rate of hate groups (i.e. stable differences between counties from different states would be ignored). Because the county-level rate of hate groups is sparsely and non-uniformly distributed across the United States, there are likely a substantial number of states with little within-state variation. Under these conditions, adding state fixed effects will increase the uncertainty (i.e. standard error) around the association between Binding values and hate group rates without adding any theoretical or interpretive value. Further, because Binding-values vary regionally (including at the state-level), adding state-level fixed effects will almost certainly partially mask the association between Binding values and hate-groups.

Finally, it is worth noting that we could not find any precedent for including state fixed effects in an analysis such as ours. For instance, Medina et al. (2018) investigate the county-level distribution of hate groups in the U.S., yet they do not include state fixed effects. Similarly, reviews of geospatial ecological regression – which is the approach we employ in this analysis – do not suggest including upper-level fixed effects (e.g. state fixed effects) when the analytical goal is to estimate a lower-level association (e.g. between counties; Lindgren & Rue, 2015; Lawson, 2013)

This all aside, we did also estimate a second model that included the standardized state-level rate of hate groups per 100k inhabitants as an additional adjustment variable. In this model, the coefficient for standardized Binding values represented the difference in the log-expected rate of hate groups *after adjusting for the rate of hate-groups in that county's state*. In other words, this model addresses the question of whether knowing a given county's Binding values provides additional information about the expected rate of hate groups if we also know the rate of hate groups in the state containing the county. Importantly, there was no substantive change in our model estimates, which suggests that even after accounting for variations in the state-level rate of hate groups, county-level Binding values explain variation in county-level hate group rates.

Breslow, N. (1982). Design and analysis of case-control studies. *Annual Review of Public Health*, 3, 29–54.

Lawson, A. B. (2013). *Bayesian disease mapping: hierarchical modeling in spatial epidemiology*. CRC press.

Lindgren, F., & Rue, H. (2015). Bayesian spatial modelling with R-INLA. *Journal of Statistical Software*, 63(19). <http://opus.bath.ac.uk/45256/>

Medina, R. M., Nicolosi, E., Brewer, S., & Linke, A. M. (2018). Geographies of Organized Hate in America: A Regional Analysis. *Annals of the Association of American Geographers. Association of American Geographers*, 108(4), 1006–1021.

Schisterman, E. F., Cole, S. R., & Platt, R. W. (2009). Overadjustment bias and unnecessary adjustment in epidemiologic studies. *Epidemiology*, 20(4), 488–495.

- 6. I am just not sure how robust these findings are. My understanding is that the authors want to demonstrate that the aggregate outcome of hate group existence is partly explained by a specific strength of some values. I understand this to mean that average local value distribution explains the outcome of hate group existence. But to show this, we would want to see that either all people moving into these regions become more racists or that some people become radicalized and much more racists. That would constitute strong evidence for such a context-effect.**

Author Response: Thank you for this comment! We fully agree that exploiting migration or relying on a number of other econometric approaches could have yielded more robust findings. Unfortunately, we do not have the data necessary for conducting such analyses. Accordingly, we see Study 2 as providing *suggestive* and *consistent* evidence regarding our hypotheses. In other words, on its own, we do not see Study 2 as providing sufficient evidence for concluding our hypotheses are supported. Indeed, this is exactly why we paired Study 2 with another observational study that relies on different data and a different level of analysis (Study 1) and with a series of controlled survey studies. Together, these studies provide convergent evidence across different levels of analysis, methods of measurement, samples, and methods of analysis. In our view, this kind of convergent evidence is exactly what is required for a finding to be robust.

Reviewers' comments:

Reviewer #3 (Remarks to the Author):

The paper is much more understandable than the initial version although it continues to be a challenging manuscript to appreciate. The improvements from the original draft, however, are extremely impressive. In many ways, I learned more from the reviewer response than from the paper itself. The radically expanded supplemental information was also quite helpful.

The authors have done a much better job in explaining MFT. [REDACTED].

Study 2 continues to be interesting and the authors have helped clarify many of the issues I was concerned about.

Studies 3-5 are solid traditional social psychology studies. Study 6, though, is compelling and fun to read. I can see the authors sitting around a table coming up with the People of the Earth vignettes. I haven't had such a good laugh in a long time. It's also a nice demonstration of very different disturbing moral challenges with similar results.

One final note. Moral Foundations Theory is not an easy framework for many people to appreciate. The foundations themselves sometimes seem arbitrary and the psychometrics are occasionally puzzling. Because the ideas underlying the theory are compelling, MFT continues to inspire many in the field. Right now, MFT is a good story but the science supporting it is still in the early stages. In some ways, the current paper is a good example. I would urge the authors to rethink the introduction a bit [REDACTED].

Reviewer #4 (Remarks to the Author):

I was a reviewer on a previous version of this MS and appreciate the changes that the authors have made in response to the last round of comments. I still have a couple concerns that I would like the authors to address:

1. Haidt's review mentions concerns about the SPLC database quality. I think the authors' response is fairly convincing but the concern may occur to other readers as well. I think the authors ought to explain the potential issues and address them, as much as they can, in the text. To be clear, I think this study adds value, and no data are perfect. Nonetheless, the limitations ought to be made clear.

2. I appreciate the addition of Study 6. I'm not convinced, though, that eating pets is the prototypical harm violation. There are definitely purity elements there as well—ironically one of Haidt's first intuitionist morality papers was subtitled "Is it wrong to eat your dog." Of course the current scenarios are different in some ways, but nevertheless some validation that pet-eating is seen as a harm violation and not a purity violation would be helpful. Along those lines, do binding values moderate the mediation between perceived immorality and approval of anti-group behavior for the pet-eating scenario. Unless I missed something, it looks likely the authors only report (non) moderation for individualizing values.

Reviewer #5 (Remarks to the Author):

Think the authors do a very good job in responding to my first two questions with respect to MrP.

But I am a little bit taken back by their response to the third point regarding state fixed effects. If there is a relationship how they propose, we would find significant effects even after including fixed effects. If the effects go away, then we learn that states with generally higher values on X also tended to have higher values on Y and that drove the relationship. I cannot see such an analysis being accepted at a top 5 journal in economics nor in political science. Our current standards have changed and we would require a more meticulous analysis of observational data.

The fact that the authors do not show the results with fixed effects let's me wonder whether the results maybe are significant after including fixed effects. Further, separating rural and urban counties would make sense. Currently, it is entirely possible that you could show a relationship between number of harvester p.c. in a county and acts of hate. Nothing is wrong with that partial correlation that you can estimate in a model - it is just far away from being causal and hence worth being explored.

In my reading the observational studies are really important since they inform us whether a mechanism, found in an online experiment, also amounts to tangible effects in the real world. In that sense, I feel that the observational studies need to be robust while it is clear that they cannot rule out every problem. What the range of problems are that authors should rule out is then a question to which each discipline has a different response which may also change over time.

I read the authors "defense" carefully and as argued above, if all factors varying with state are orthogonal to their measures, we do not need to include fixed effects. But if we do, the results would not change. In all other cases one has to include them (check e.g. Mundlack's work on using means of groups to block bias from the 1970s). I assume these are differences that arise from different disciplinary backgrounds and this makes it ultimately an editorial decision. I can only repeat myself and say that in leading economics, political science, or sociology journal such an analysis would most likely not stand.

November 18, 2020

Dear Reviewers,

We would also like to thank you for offering us the opportunity to continue improving our manuscript, and providing us with such constructive comments and helpful suggestions.

We have carefully revised our paper and have addressed all the issues discussed by the reviewers. Below we quote the entirety of the reviewers' comments (numbered and in bold) and detail how we modified the paper to respond to each point (in italic).

RESPONSES TO REVIEWER #3:

- 1. The paper is much more understandable than the initial version although it continues to be a challenging manuscript to appreciate. The improvements from the original draft, however, are extremely impressive. In many ways, I learned more from the reviewer response than from the paper itself. The radically expanded supplemental information was also quite helpful.**

Author Response: *We thank the Reviewer for positive and constructive feedback. We also agree that the paper has improved significantly thanks to the reviews in the previous rounds.*

- 2. The authors have done a much better job in explaining MFT. [REDACTED]**

Author Response: [REDACTED]

3. **Study 2 continues to be interesting and the authors have helped clarify many of the issues I was concerned about.**

Author Response: *We thank the Reviewer for positive feedback. We are happy that our revision has made the manuscript clearer.*

4. **Studies 3-5 are solid traditional social psychology studies. Study 6, though, is compelling and fun to read. I can see the authors sitting around a table coming up with the People of the Earth vignettes. I haven't had such a good laugh in a long time. It's also a nice demonstration of very different disturbing moral challenges with similar results.**

Author Response: *We thank the Reviewer for this comment. As mentioned, Studies 3-5 were social psychological studies designed to better understand the mechanisms that explain the link between moral values and hateful acts. Study 6 is a more recent addition, in which we have specific manipulations so we can better study the differential effects in the Binding and Individualizing moral domains.*

5. **One final note. Moral Foundations Theory is not an easy framework for many people to appreciate. The foundations themselves sometimes seem arbitrary and the psychometrics are occasionally puzzling. Because the ideas underlying the theory are compelling, MFT continues to inspire many in the field. Right now, MFT is a good story but the science supporting it is still in the early stages. In some ways, the current paper is a good example. I would urge the authors to rethink the introduction a bit [REDACTED].**

Author Response: *We agree with the Reviewer that MFT may not be immediately accessible for some readers. We also completely agree that "the science supporting [MFT] is still in the early stages." Hopefully, as the Reviewer points out, this empirical paper can be used to refine the theoretical predictions of the theory. We have now added further information about MFT in our Introduction. We hope that the changes to the Introduction can better present our theoretical framework. Our rationale for presenting our [REDACTED] geo-spatial analyses first was to 1. start the paper with observational studies, before discussing the experimental work 2. present and analyze large but noisy datasets and prepare the readership for consequent studies which dive into individual-level mechanisms. [REDACTED]*

RESPONSES TO REVIEWER #4:

1. **I was a reviewer on a previous version of this MS and appreciate the changes that the authors have made in response to the last round of comments. I still have a couple concerns that I would like the authors to address:**

Author Response: *We appreciate the positive and constructive feedback, in addition to the continued engagement with our manuscript.*

- 2. Haidt’s review mentions concerns about the SPLC database quality. I think the authors’ response is fairly convincing but the concern may occur to other readers as well. I think the authors ought to explain the potential issues and address them, as much as they can, in the text. To be clear, I think this study adds value, and no data are perfect. Nonetheless, the limitations ought to be made clear.**

Author Response: *We thank the Reviewer for raising this important point. We completely agree that there are potential issues with SPLC data. However, after thoroughly researching the field, we consider this dataset to be the best available data for testing our hypothesis, as described in detail in our response to Dr. Haidt. As suggested, we have now specifically added the following paragraph to our manuscript: “Of note, the SPLC data is not without limitations. The hate group activities are collected and indexed in their database by volunteers and staffers, hence they may be subject to organizational and personal biases (Moser, 2019). Although SPLC conducts extensive data collection efforts, there is no conclusive evidence that the data are complete or without bias. Recent research has found positive associations between regional variations in Uniform Crime Reporting (compiled by the FBI) and the geographic distribution SPLC-identified hate group activities (Jendryke & McClure, 2019).”*

- 3. I appreciate the addition of Study 6. I’m not convinced, though, that eating pets is the prototypical harm violation. There are definitely purity elements there as well—ironically one of Haidt’s first intuitionist morality papers was subtitled “Is it wrong to eat your dog.” Of course the current scenarios are different in some ways, but nevertheless some validation that pet-eating is seen as a harm violation and not a purity violation would be helpful.**

Author Response: *Thank you for this comment and suggestion. We agree that it is certainly possible – if not probable – that the “eating pets” condition contains elements of purity violations. However, while we considered more prototypical harm violations, such as animal abuse, we wanted to avoid behaviors that were illegal and thus would introduce a new confounding factor. That is, one aim of our study design was to select Individualizing and Binding violations that were not illegal.*

Nonetheless, to directly address this concern, we have added a new validation analysis where we demonstrate the following:

- 1. Individualizing values moderate the perceived moral wrongness of the People of the Earth eating pets more strongly than Binding values do.*
- 2. Individualizing values moderate the perceived moral wrongness of the eating pets condition more strongly than Individualizing values moderate the perceived moral wrongness of the taboo sex condition.*
- 3. Binding values moderate the perceived moral wrongness of the taboo sex rituals condition more strongly than Individualizing values do.*
- 4. Binding values moderate the perceived moral wrongness of the taboo sex rituals condition more strongly than Binding values moderate the perceived moral wrongness of the eating pets condition.*

Again, while we agree that it is certainly possible that the “eating pets” condition is also related to purity concerns, these results suggest that our manipulations of Individualizing and Binding violations were effective and distinct.

- 4. Along those lines, do binding values moderate the mediation between perceived immorality and approval of anti-group behavior for the pet-eating scenario. Unless I missed something, it looks likely the authors only report (non) moderation for individualizing values.**

Author Response: *Thank you for this question. In our revised manuscript we now include a new analysis investigating whether Binding values moderate the mediation effect of perceived moral wrongness in the domain of Individualizing values. We expected this analysis to reveal a null to weak moderation effect, because, as discussed above, it’s possible that our Individualizing Values violation manipulation may also partially function as a weak Binding values violation.*

As expected, while we did observe a small moderation effect, this effect was substantially smaller than the analogous effect in the Binding values violation condition.

RESPONSES TO REVIEWER #5:

- 1. Think the authors do a very good job in responding to my first two questions with respect to MrP.**

Author Response: *We thank the reviewer for their helpful comments and for the continued engagement with our paper.*

- 2. But I am a little bit taken back by their response to the third point regarding state fixed effects. If there is a relationship how they propose, we would find significant effects even after including fixed effects. If the effects go away, then we learn that states with generally higher values on X also tended to have higher values on Y and that drove the relationship. I cannot see such an analysis being accepted at a top 5 journal in economics nor in political science. Our current standards have changed and we would require a more meticulous analysis of observational data.**

Author Response: *We thank the reviewer for this comment, as it demonstrates that they have deeply engaged with our manuscript and response, and we are very grateful for that. However, we are a little surprised by the reviewer’s surprise. We feel that the arguments we presented in our previous response were uncontroversial.*

We continue to see two potential issues – one statistical and one theoretical – with using a state-level fixed effects estimator to estimate the county-level association between county Binding values and hate group rates. Perhaps we could have explained these more clearly in our previous response and, if that is the case, we apologize for the lack of clarity.

The statistical issue we refer to is simply the fact that fixed-effects estimators tend to reduce statistical power. This has been common knowledge for some time; indeed, econometricians Angrist and Pischke (2008) described this issue as “throwing the baby out with the bathwater” more than 10 years ago :

Although they control for a certain type of omitted variable, fixed effects estimates are notoriously susceptible to attenuation bias from measurement error. [...] This fact may account for smaller fixed effects estimates. [...] A variant on the measurement error problem in panel data arises from that fact that the differencing and deviations from means estimators used to control for fixed effects typically remove both good and bad variation. In other words, these transformations may kill some of the omitted variables bias bathwater, but they also remove much of the useful information in the baby, the variable of interest. [...] At a minimum, therefore, it’s important to avoid overly strong claims when interpreting fixed effects estimates [...].

*For more current treatments of the biases that can be induced by fixed-effects estimators, Plumper and Troeger (2019) offer a thorough analysis that was published in *Political Analysis*, a leading political science journal, and (potentially) the authoritative methods venue in that field. Hill et al. (2020), despite not being political scientists nor economists, also offer a clear, though high-level, discussion of these matters.*

The issue of loss of power and sensitivity to levels of within-group variation is particularly important for our second study, because we have good reason to expect relatively low within-state variation of both hate group rates and Binding values. We expect the former because hate groups are relatively rare and we expect the latter because we relied on MrP to estimate county-level Binding values and those county-level estimates will be smoothed toward state-level means (note, this is a desirable feature of MrP that helps mitigate problems caused by sparsity). As such, we can infer a priori that a state-level fixed effects estimator will have lower power or, more specifically, that it will attenuate the point estimate and amplify the standard error of the parameter of interest. This makes null results difficult to interpret: they could imply the presence of unmeasured confounds, but they could also merely imply insufficient statistical power.

*In short, we agree that if including state fixed effects makes “the effects go away” then we can conclude that states with generally higher values on *X* also tend to have higher values on *Y*. However, this conclusion is inconclusive, as there are a number of potential explanations for this result. This is why Angrist and Pischke, in their canonical introduction to econometrics, stress the importance of interpreting estimates derived from fixed effects estimators with caution.*

Aside from this statistical issue with adjusting for state fixed effects, we also remain unconvinced that states are a natural and necessary grouping variable in this particular domain. Most classical treatments of fixed effects estimators focus on longitudinal panel data. In such cases, measurements are nested within a unit, such as a state, country, company, or individual, and fixed effects adjustments, in a loose sense, yield a model that lets units serve as their own controls. However, using state-level fixed effects to adjust county-level estimates has substantively different implications. The validity of that approach rests on the assumption that states are a meaningful grouping variable for counties. In many contexts, such as analyses of variables that are meaningfully driven by state-level differences, this certainly can be a valid assumption. However, this does not mean that it is always a valid assumption. Further, in the domain of Binding values and hate group rates, it is not clear why states would be a natural grouping variable that must be adjusted for, especially when known risk factors are already adjusted for.

In sum, our issue with state fixed effects is that they will decrease statistical power and it is not at all clear what unmeasured state-level mechanisms would be the real factors driving the county-level effects

that we observe. Finally, we would like to repeat that these are not controversial issues. They are well known among methodologists from economics, political science, and even some of the other social sciences.

Angrist, J. D., & Pischke, J. S. (2008). *Mostly harmless econometrics: An empiricist's companion*. Princeton university press.

Hill, T. D., Davis, A. P., Roos, J. M., & French, M. T. (2020). Limitations of fixed-effects models for panel data. *Sociological Perspectives*, 63(3), 357-369.

Plümper, T., & Troeger, V. E. (2019). Not so harmless after all: The fixed-effects model. *Political Analysis*, 27(1), 21-45.

3. The fact that the authors do not show the results with fixed effects let's me wonder whether the results maybe are significant after including fixed effects.

Author Response: *We note above that our dependent variable and independent variable of interest are likely to have limited within-state variance. One consequence of this is that a state fixed effects estimator is likely to be underpowered and thus inconclusive. However, though we remain unsure of what unmeasured state-level confounds would be a better explanation for the association we observe, this does not mean that we are not sensitive to the issue of unmeasured state-level confounds. We simply do not believe that our data are well suited for addressing that issue.*

Despite this, we have added a new section to our Supplemental Materials where we discuss and report estimates from a model with fixed effects estimates. In this model, the association between county-level hate group rates and county-level Binding values is no longer distinguishable from null. Specifically, the point estimate of the effect attenuated toward zero (0.19 vs 0.32) and it's SE was amplified (.18 vs .10). While these results are consistent with the explanation that state-level associations were responsible for driving the effect we observed, they are also consistent with the explanation that there is insufficient within-state variation to estimate the effect with precision. This is exactly the kind of inconclusive situation we sought to describe in our previous response and in our discussion above.

4. Further, separating rural and urban counties would make sense. Currently, it is entirely possible that you could show a relationship between number of harvester p.c. in a county and acts of hate. Nothing is wrong with that partial correlation that you can estimate in a model - it is just far away from being causal and hence worth being explored.

Author Response: *Thank you for this comment. However, we are not entirely sure what you mean by separating rural and urban counties. Most likely, it seems that you are suggesting estimating separate models for urban and rural counties. We agree that this would be interesting, however it seems likely that the implications of this model might be hard to interpret, given the associations between rural and urban counties and other factors including our IV of interest, Binding values. That is, for instance, we know that more rural areas tend to be higher on Binding values (and, indeed, related factors such as the culture of honor) compared to urban areas. Accordingly, one consequence of this decomposition would be reducing variance in ways that could lead to findings that are difficult to interpret.*

That said, we do think this is an interesting question. However, we feel it is currently outside the scope of this project and will be better left for future work.

5. **In my reading the observational studies are really important since they inform us whether a mechanism, found in an online experiment, also amounts to tangible effects in the real world. In that sense, I feel that the observational studies need to be robust while it is clear that they cannot rule out every problem. What the range of problems are that authors should rule out is then a question to which each discipline has a different response which may also change over time.**

Author Response: *Thank you for this comment. We fully agree and, indeed, this question whether effects observed in experiments can be observed in the real world played a core role in our study design process.*

We also fully agree that it is key that our observational studies need to be robust. In both observational studies, we took steps that are well beyond the norm, such as [REDACTED] using MrsP to obtain the best estimates possible given our data, and adjusting for spatial autocorrelation in our final models. We would like to note that this issue of spatial autocorrelation has often been overlooked not only in psychology, but also political science and economics. At every step, we have sought to make our analyses as robust as possible.

Further, in our previous revision, we sought to address as many of the issues raised by reviewers as possible. For example, per the reviewer's suggestion, we added an extensive validation study designed to evaluate the likely validity of our MrsP estimates.

Ultimately, it seems that we share many perspectives on the current work, with the one issue of contention being the necessity of state fixed effects. Due to the reasons discussed above, we simply don't believe our data are well suited for ruling out the possibility of state-level confounds.

We would also like to note that we would never seek to publish a study like our second study on its own. Indeed, we generally find cross-sectional studies suggestive at best and it is not the case that we are expecting Study 2 to stand on its own merit. As we note in our manuscript, studying the psychological mechanisms involved in acts of hate is very difficult; to address this difficulty, we have designed and conducted a set of complementary studies, where the goal was to make up for the weaknesses of one design with the strengths of another. In total, six rigorous studies point to the same effect, and that is the package that we are aiming to publish, not Study 2 on its own.

I read the authors "defense" carefully and as argued above, if all factors varying with state are orthogonal to their measures, we do not need to include fixed effects. But if we do, the results would not change. In all other cases one has to include them (check e.g. Mundlack's work on using means of groups to block bias from the 1970s). I assume these are differences that arise from different disciplinary backgrounds and this makes it ultimately an editorial decision. I can only repeat myself and say that in leading economics, political science, or sociology journal such an analysis would most likely not stand.

Author Response: *We thank the reviewer again, for engaging with this issue. We are well versed with the problem of confounding between- and within-group variance and we agree that estimates that do not decompose or adjust for these sources of variation will be biased. However, again, fixed effects estimators are not a silver bullet. In our previous response and above we explain when and why fixed effects estimators can introduce bias. We have also cited discussions of this issue that were published in a leading political science journal and written by leading econometricians. To anyone who believes that*

fixed effects estimators offer a final answer or perfect solution to the problem of between- and within-group confounding, we would strongly suggest reviewing this literature. That said, we would not expect any cross-sectional analysis – whether or not it adjusts for fixed effects – to stand on its own. Across the social sciences we have noticed a recent movement toward stronger identification strategies and, for a study to stand on its own, we believe that cross-sectional designs are no longer satisfactory. However, again, we never intended for our second study to stand on its own. We conducted the most rigorous study that we could, given the limitations imposed by data availability. To address these limitations, we conducted 5 additional studies that address the question of interest from a variety of perspectives. All six, even though complementary, rely on different types of data, and use different populations. Nonetheless, all our studies point to the same mechanism. We feel that it would be very strange to dismiss six studies that demonstrate convergent findings simply because one study suffered the limitation of low within-state variation. Indeed, it would be hard to see that as anything other than “throwing the baby out with the bathwater”, as argued by Angrist and Pischke (2008).

Again, we fully agree that Study 2 should not stand on its own, no matter what the discipline. This is exactly why we have our geo-spatial analysis complemented with a study using state of the art natural language processing and three behavioral studies, two of which use stratified representative samples. Frankly, we do not know of any study in leading economics, political science, sociology or psychology journals which provide such diversification of methodologies and data to address complex socio-psychological phenomena.

We are providing access to all relevant code and publicly available data in our Open Science Foundation repository, available here: <https://osf.io/67cdg/>.

Thank you again for your time and consideration.

Reviewers' comments:

Reviewer #3 (Remarks to the Author):

This is the third time I have reviewed the paper. The authors have been responsive to most of my comments. [REDACTED]. Perhaps because of the review process itself, there is a sense that every new revision of the paper becomes more technical. Fortunately, the discussion section will help readers understand the logic and findings of the project.

Reviewer #4 (Remarks to the Author):

I was asked to comment on two aspects of the current MS:

1. Whether the current version is responsive to my previous review.
2. Whether the authors' treatment of state-level fixed effects is correct/appropriate.

On the first point, I'm happy with their changes in response to my previous review.

On the second: I am not an expert on econometrics. But it seems to me that the non-significant effects when state-level fixed effects are included could be due to some unmeasured correlate of state confounding the results. Or, it could be because including these fixed effects reduces power. There's just no way to know for sure which of these is the right explanation (or, in fact, both things could be going on).

Given this uncertainty, I think the authors need to address this question more thoroughly in the main text of the MS, so that readers can judge for themselves. I don't think that this issue is fatal to the paper. But I do think it deserves more prominent examination, given that many readers may have a similar reaction to Reviewer 5.

Reviewer #5 (Remarks to the Author):

The one point of contention that remains is whether fixed effects need to be added or not. The results adding them show that there is no significant relationship.

As I wrote in my earlier response, for some disciplines it seems obvious that a model needs to have FE when the paper goes for a top outlet. The authors argue that there are problems and cite Angrist and Pischke.

The same authors have an entire chapter (#5) on fixed effects and why this is such a popular strategy to deal with unobserved confounding factors. The selective choice of one paragraph highlighting that there may be a problem as if this is the main claim on fixed effects by the authors is surprising. Importantly, the authors then on page 226 argue that there are remedies but none of their proposals include dropping fixed effects!

Plumper and Troeger are looking at a specific deficiency of FE estimators given that "...dynamic misspecification and omitted within variation correlated with one of the regressors" — I do not see that the authors argue that they suffer from these specific problems.

But as written in my last review — I think that this is an editorial matter. Editors need to decide how high they set the “burden of proof”.

Reviewer #6 (Remarks to the Author):

This is my first time reviewing the paper (though I have read the authors' responses to previous reviews). I have been asked to comment specifically on the intergroup aspects of the piece as well as the state fixed effects discussion that is ongoing between the authors and two of the reviewers.

The second issue is fairly straightforward to my mind, so I'll start there. As someone who has published with both psychologists and economists I can attest to the different standard practices between the two fields. Because this is a top interdisciplinary journal I would advise the authors to publish either the model that includes state fixed effects or both models. There is no utility in omitting the results with state fixed effects. People who would be skeptical of the results due to the difference in how the models are specified will be just as, if not more skeptical if the state fixed effects specification is missing! It also raises interesting questions and ideas for future research to try to tease apart why the different specifications are yielding different results. It's more beneficial for everyone—authors and readers alike—for both to appear in the paper (either both in the main text or with one in the supplementary materials).

As to the intergroup aspects of the paper the way I've interpreted the authors' model is that differences between moral values (specifically those related to group preservation) highlights group boundaries, which in turn activates out-group threat, which in turn foments hate crimes against that out-group. At a high level this fits with lots of intergroup theorizing that out-group threat activates in-group cohesion to facilitate defense against that threat. There is, however, some circularity in this argument when you start to scratch the surface--it isn't clear whether perceptions of differences in values about group variation highlight intergroup boundaries which then drives hate crimes (e.g., “wow they think really differently from us about what makes a group a group, which we find very threatening and different from ourselves, so it's okay to harm them”) or whether it's the other way around (e.g., we don't like them for other reasons (see recent COVID-related anti-Asian hate crimes); because we don't like them it's clear that we're two different groups, and because we're so different there's no way they think about groups the same way that we do so it's acceptable to harm them”). Or maybe it's a third model in which binding values are a moderator of the relationship between out-group threat (which can arise for many reasons) and hate crimes. But then is that perpetrator-specific moderator (i.e., groups that center binding values) also interacting with a victim-specific moderator (i.e., groups that *don't* center binding values)? Or as in studies 3-4 is the victim-wrongness built into the intergroup threat that is the IV in this scenario (e.g., economic threat from Mexican people, which isn't really a moral threat at all)? Or based on [REDACTED] study 2, which document a relationship between binding language and hate speech/groups, are binding morals serving the function of a mediator? What I'm ultimately getting at is that the introduction needs to be revised to articulate better precisely which model the authors are testing (and consider adding a figure to explain how all of these constructs relate to one another). Right now the paper is a collection of interesting findings but it's very hard to back out how all the pieces fit together.

There's another complicating factor which is that the authors claim that hate crimes are associated specifically with right wing ideology. Again, that's what [REDACTED] Study 2 partially indicate (if we use binding values as proxies for right-wing ideology) but it further

narrows the model because it suggests that the “Moralized Threat Hypothesis” should actually be referred to as the “Binding Values Threat Hypothesis.” This isn’t a model about general group dynamics or even group-based morality (which can center on fairness and harm avoidance, e.g., as in Black Lives Matter), it’s a model about those groups who specifically coalesce around loyalty, authority, and purity.

March 14, 2021

Dear Reviewers,

We would like to thank all six reviewers for their continued support and constructive help with the revision of our manuscript. We have carefully revised our paper and have addressed all the issues discussed by the reviewers. Below we quote the entirety of the reviewers' comments (in italic) and detail how we modified the manuscript to respond to each point.

Reviewer #3 (Remarks to the Author):

This is the third time I have reviewed the paper. The authors have been responsive to most of my comments. [REDACTED]

Perhaps because of the review process itself, there is a sense that every new revision of the paper becomes more technical. Fortunately, the discussion section will help readers understand the logic and findings of the project.

We would like to thank the Reviewer for their continued support. We are glad that the Reviewer seems to be overall satisfied with the state of the paper.

Reviewer #4 (Remarks to the Author):

I was asked to comment on two aspects of the current MS:

- 1. Whether the current version is responsive to my previous review.*
- 2. Whether the authors' treatment of state-level fixed effects is correct/appropriate.*

On the first point, I'm happy with their changes in response to my previous review.

Thank you again for all your help and the time you spent on our manuscript. We are very happy that you are happy with our changes in response to your review.

On the second: I am not an expert on econometrics. But it seems to me that the non-significant effects when state-level fixed effects are included could be due to some unmeasured correlate of state confounding the results. Or, it could be because including these fixed effects reduces power. There's just no way to know for sure which of these is the right explanation (or, in fact, both things could be going on).

Given this uncertainty, I think the authors need to address this question more thoroughly in the main text of the MS, so that readers can judge for themselves. I don't think that this issue is fatal to the paper. But I do think it deserves more prominent examination, given that many readers may have a similar reaction to Reviewer 5.

Thank you for this comment. We fully agree with the Reviewer. We did this. We have now included the analysis with fixed effects in the main document.

Reviewer #5 (Remarks to the Author):

The one point of contention that remains is whether fixed effects need to be added or not. The results adding them show that there is no significant relationship.

As I wrote in my earlier response, for some disciplines it seems obvious that a model needs to have FE when the paper goes for a top outlet. The authors argue that there are problems and cite Angrist and Pischke.

The same authors have an entire chapter (#5) on fixed effects and why this is such a popular strategy to deal with unobserved confounding factors. The selective choice of one paragraph highlighting that there may be a problem as if this is the main claim on fixed effects by the authors is surprising. Importantly, the authors then on page 226 argue that there are remedies but none of their proposals include dropping fixed effects!

Plumper and Troeger are looking at a specific deficiency of FE estimators given that "...dynamic misspecification and omitted within variation correlated with one of the regressors" — I do not see that the authors argue that they suffer from these specific problems.

But as written in my last review — I think that this is an editorial matter. Editors need to decide how high they set the "burden of proof".

We thank the Reviewer for their comments and the time spent on our work. We seem to disagree about whether or not the state fixed-effects should be included in the models. We do agree, however, that there are differences across disciplines about this issue. Therefore, as Reviewer 4 and 6 suggested, we have now included both models in the main document allowing the reader to decide which model and results to trust. We would also like to highlight that this paper reports on a multi-methodological line of research examining moral values and extreme behavioral expressions of prejudice. Therefore, our studies should be regarded as complementary. Again, we thank the Reviewer for their helpful comments through the revision process.

Reviewer #6 (Remarks to the Author):

This is my first time reviewing the paper (though I have read the authors' responses to previous reviews). I have been asked to comment specifically on the intergroup aspects of the piece as well as the state fixed effects discussion that is ongoing between the authors and two of the reviewers.

We thank the reviewer for positive feedback and constructive comments.

The second issue is fairly straightforward to my mind, so I'll start there. As someone who has published with both psychologists and economists I can attest to the different standard practices between the two fields. Because this is a top interdisciplinary journal I would advise the authors to publish either the model that includes state fixed effects or both models. There is no utility in omitting the results with state fixed effects. People who would be skeptical of the results due to the difference in how the models are specified will be just as, if not more skeptical if the state fixed effects specification is missing! It also

raises interesting questions and ideas for future research to try to tease apart why the different specifications are yielding different results. It's more beneficial for everyone—authors and readers alike—for both to appear in the paper (either both in the main text or with one in the supplementary materials).

We thank the reviewer for this comment. We completely agree with this and it resonates with our experience working across disciplines. As suggested, we now fully report the fixed effects model in the manuscript. We also thank the reviewer for highlighting the important fact that our results will have interesting methodological implications and will raise novel questions for future research.

*As to the intergroup aspects of the paper the way I've interpreted the authors' model is that differences between moral values (specifically those related to group preservation) highlights group boundaries, which in turn activates out-group threat, which in turn foments hate crimes against that out-group. At a high level this fits with lots of intergroup theorizing that out-group threat activates in-group cohesion to facilitate defense against that threat. There is, however, some circularity in this argument when you start to scratch the surface--it isn't clear whether perceptions of differences in values about group variation highlight intergroup boundaries which then drives hate crimes (e.g., "wow they think really differently from us about what makes a group a group, which we find very threatening and different from ourselves, so it's okay to harm them") or whether it's the other way around (e.g., we don't like them for other reasons (see recent COVID-related anti-Asian hate crimes); because we don't like them it's clear that we're two different groups, and because we're so different there's no way they think about groups the same way that we do so it's acceptable to harm them"). Or maybe it's a third model in which binding values are a moderator of the relationship between out-group threat (which can arise for many reasons) and hate crimes. But then is that perpetrator-specific moderator (i.e., groups that center binding values) also interacting with a victim-specific moderator (i.e., groups that *don't* center binding values)? Or as in studies 3-4 is the victim-wrongness built into the intergroup threat that is the IV in this scenario (e.g., economic threat from Mexican people, which isn't really a moral threat at all)? [REDACTED] Study 2, which document a relationship between binding language and hate speech/groups, are binding morals serving the function of a mediator? What I'm ultimately getting at is that the introduction needs to be revised to articulate better precisely which model the authors are testing (and consider adding a figure to explain how all of these constructs relate to one another). Right now the paper is a collection of interesting findings but it's very hard to back out how all the pieces fit together.*

We thank the reviewer for thoughtful consideration of our findings and helpful suggestions. We generally agree with the reviewer that fleshing out alternative theoretical models can strengthen the paper and situate it better within the broader intergroup literature. We would also like to point out that Nature Communications has a strict word limit for the Introduction and General Discussion, so it may not be practical to summarize different approaches and theoretical frameworks at length. Here, we rely on Moral Foundations Theory which is an extension of the Social Intuitionist Model (Haidt, 2001), suggesting that moral reasoning and judgment come *after* a gut-level feeling of like or dislike. We do not test this assumption here as it has been tested and established elsewhere, but it helps clarify our theoretical assumptions about outgroup hate. Hence, theoretically, MFT would favor your second argument: "because we don't like them, it's clear that we're two different groups, and because we're so different there's no way they think about groups the same way that we do so it's acceptable to harm them."

To clarify our process model, we added a schematic to the main text. This process model depicts a causal chain linking moral values to outgroup moral violation to perceived moral wrongness and finally to perceived justification of EBEPs. In our observational results (Studies 1 and 2), the relationship between moral values and hate speech/hate groups is established. Then we turn to controlled behavioral experiments to dissect the processes. We explain in the text that our social psychological studies test the

Moralized Threat Hypothesis, an extension of the threat hypothesis. Across these different studies, we rely on real-world outgroups (Muslims and undocumented Mexican immigrants), a fictional subgroup with perceived threat (Sandirians), and fictional groups without perceived threats (People of the Earth). The fact that we replicated our results across such different social groups speaks to strong evidence for the core prediction of the Moralized Threat Hypothesis.

In sum, as suggested by the reviewer, we added a figure to precisely articulate our process model. We experimentally test different parts of the model with a diverse number of outgroups. We share the sentiment with the reviewer that “at a high level [our theoretical model] fits with lots of intergroup theorizing that out-group threat activates in-group cohesion to facilitate defense against that threat.” Our refined process model suggests that moral values, specifically binding values, lead to an increase in perceived outgroup moral violation and wrongness, which in turn leads to higher likelihood of perceived justification of extreme behavioral expressions of prejudice. We hope that the present results contribute to the cumulative and emerging science of intergroup hate and prejudice from a moral psychological perspective.

There’s another complicating factor which is that the authors claim that hate crimes are associated specifically with right wing ideology. Again, [REDACTED] Study 2 partially indicate (if we use binding values as proxies for right-wing ideology) but it further narrows the model because it suggests that the “Moralized Threat Hypothesis” should actually be referred to as the “Binding Values Threat Hypothesis.” This isn’t a model about general group dynamics or even group-based morality (which can center on fairness and harm avoidance, e.g., as in Black Lives Matter), it’s a model about those groups who specifically coalesce around loyalty, authority, and purity.

We thank the reviewer for this comment. While we acknowledge in the Introduction and the General Discussion [REDACTED], we adjust for political ideology in Studies 2-5. [REDACTED]. Our findings in Study 2 suggest that binding values held in a county are related to the presence of hate groups in that county after adjusting for county-level political ideology. Our models are shown to be robust to political ideology in Studies 3-5 (please see Supplemental Materials). Finally, to test whether the moral violations in Studies 3-5 could potentially be understood as existential (Study 3) or economic threats (Studies 4 & 5), we designed Study 6. In Study 6, we also experimentally tested if this pattern of results is unique to the binding foundations. We are cautious about our wording in the manuscript: “Our findings tentatively suggest that Binding values are a more severe risk factor for acts of hate, relative to Individualizing values.”

In addition, a recent meta-analysis (Kivikangas et al., 2020) shows that binding values may not be as strongly associated with right-wing political ideology as previously thought, especially in diverse samples (our Study 6 used a nationally stratified sample to represent a cross-section of U.S. adults in terms of age, gender, ethnicity, and political ideology). So, we think the reviewer’s point, that our results rely on the assumption that “binding values [are] proxies for right-wing ideology,” is unwarranted, especially when political ideology is explicitly accounted for. In terms of our wording in “group-based morality”, we would like to point out that this terminology is borrowed from MFT. We would also like to point out that in the MFT framework, social-justice concerns (as in the BLM movement) may not be considered “group-based” or “binding” morality, as they are concerned with the well-being of individual group members, not the group itself.

Based on these points above, we decided to keep the name of our model as it was in previous rounds of revision (the Moralized Threat Hypothesis). We do agree with the reviewer that most of our results point

to the primacy of binding (vs. individualizing) values, but that does not nullify the validity of the name of our model since binding values *are* “moral” values versus, for example, social conventions. Our Study 6 provides evidence that our model is NOT “a model about those groups who specifically coalesce around loyalty, authority, and purity,” but we have added clarification on constraints on the generality of these findings to groups who specifically coalesce around loyalty, authority, and purity in our General Discussion.

We would like to thank all the six reviewers once again for their continued help and support throughout this long process. We are confident that we have addressed all the reviewer’s concerns in this round, and in the previous rounds of revisions.

Reviewers' comments:

Reviewer #6 (Remarks to the Author):

The authors have addressed my concerns and I'm excited to see this work in press. I especially appreciate their attentiveness to an 11th hour reviewer!

May 17, 2021

Dear Reviewers,

We would like to thank all the reviewers for their help with the revision of our manuscript. Below we quote the entirety of the reviewers' comments (in italic) and detail how we modified the manuscript to respond to each point.

Reviewer #6 (Remarks to the Author):

The authors have addressed my concerns and I'm excited to see this work in press. I especially appreciate their attentiveness to an 11th hour reviewer!

We thank the reviewer for such prompt response to our revision, and for providing positive feedback and constructive comments.

We would like to thank all the six reviewers once again for their continued help and support throughout this long process. We are confident that we have addressed all the reviewer's concerns in this round, and in the previous rounds of revisions.